# Shedding of *N*-acetylglucosaminyltransferase-V is regulated by maturity of cellular *N*-glycan

Tetsuya Hirata[1], Misaki Takata[2], Yuko Tokoro[1], Miyako Nakano[2] & Yasuhiko Kizuka [1✉]

The number of *N*-glycan branches on glycoproteins is closely related to the development and aggravation of various diseases. Dysregulated formation of the branch produced by *N*-acetylglucosaminyltransferase-V (GnT-V, also called as MGAT5) promotes cancer growth and malignancy. However, it is largely unknown how the activity of GnT-V in cells is regulated. Here, we discover that the activity of GnT-V in cells is selectively upregulated by changing cellular *N*-glycans from mature to immature forms. Our glycomic analysis further shows that loss of terminal modifications of *N*-glycans resulted in an increase in the amount of the GnT-V-produced branch. Mechanistically, shedding (cleavage and extracellular secretion) of GnT-V mediated by signal peptide peptidase-like 3 (SPPL3) protease is greatly inhibited by blocking maturation of cellular *N*-glycans, resulting in an increased level of GnT-V protein in cells. Alteration of cellular *N*-glycans hardly impairs expression or localization of SPPL3; instead, SPPL3-mediated shedding of GnT-V is shown to be regulated by *N*-glycans on GnT-V, suggesting that the level of GnT-V cleavage is regulated by its own *N*-glycan structures. These findings shed light on a mechanism of secretion-based regulation of GnT-V activity.

[1] Institute for Glyco-core Research (iGCORE), Gifu University, Gifu 501-1193, Japan. [2] Graduate School of Integrated Sciences for Life, Hiroshima University, Higashihiroshima 739-8530, Japan. ✉email: kizuka@gifu-u.ac.jp

Protein functions are strictly regulated by post-translational modifications, including phosphorylation, methylation, lipidation, and glycosylation. Most proteins in the secretory pathway undergo glycosylation. Asparagine-linked glycosylation (N-glycosylation), a major type of glycosylation, is well conserved among various organisms and is essential for protein folding, trafficking, activity, and signal transduction[1–4]. Alterations in N-glycan structures can cause disease development and aggravation, for example in Alzheimer's disease, diabetes, and cancer[5–8]. Therefore, it is pivotal to understand how N-glycan biosynthesis is regulated in cells and how it is dysregulated in diseases.

Biosynthesis of N-glycans starts in the endoplasmic reticulum (ER), and a common precursor oligosaccharide consisting of 14 sugar residues is transferred *en bloc* to the Asn residue of an N–X–S/T consensus sequence (where X is any amino acid except Pro)[3]. N-Glycosylated proteins are then transported to the Golgi after trimming of three glucose (and one mannose) residues, and N-glycans are further modified by various glycosidases and glycosyltransferases. This process converts immature oligomannose-type N-glycans to the mature complex-type N-glycans, which have varying numbers of branches (Fig. 1a). N-Acetylglucosaminyltransferase (GnT)-III[9], -IV[10], and -V[6,11] transfer GlcNAc to β-Man via a β1,4-linkage, to α1,3-Man via a β1,4-linkage, and to α1,6-Man via a β1,6-linkage, respectively (Fig. 1a). In addition to these GlcNAc branches, a fucose residue can be attached to the innermost GlcNAc residue by fucosyltransferase 8 (FUT8)[12,13] (Fig. 1a).

These branched structures of N-glycans have close relationships to specific diseases. For example, bisecting GlcNAc, which is the GlcNAc branch synthesized by GnT-III, is involved in Alzheimer's disease because the loss of GnT-III resulted in both the reduction of amyloid-β deposition and improvement of cognitive function in Alzheimer's disease mouse models, owing to the impaired function of amyloid-β-producing enzyme BACE1[14,15]. Knockout (KO) mice of another GlcNAc-branching enzyme—GnT-IVa—exhibited a type 2 diabetes-like phenotype because of the impaired function of glucose transporter 2 in pancreatic β-cells[16]. The β1,6-GlcNAc branch synthesized by GnT-V is involved in cancer development and malignancy as tumor growth and metastasis were greatly inhibited in GnT-V-KO mice[17]. Mechanistically, galectin-3-mediated retention of epidermal growth factor receptor at the cell surface and downstream signaling were reduced in GnT-V-KO cells[18]. Moreover, knocking down GnT-V inhibited cell migration and invasion, probably because of the reduced β1,6-GlcNAc level on N-cadherin[19]. These observations indicate that regulation of the biosynthesis of N-glycan branches is a potential therapeutic approach to these diseases. Therefore, it is of critical importance to fully understand the regulatory mechanisms of the glycosyltransferases involved in N-glycan branching.

Several regulatory factors of glycosyltransferase activities have been reported, such as subcellular localization[20–23] and formation of enzyme complexes[24–26]. These post-transcriptional mechanisms greatly impact glycosyltransferase activity and the resultant glycan profiles in cells. Furthermore, cellular activities of glycosyltransferases for N-glycan biosynthesis were shown to mutually affect each other by compensatory or feedback mechanisms. For example, loss of bisecting GlcNAc synthesized by GnT-III resulted in elevation of many terminal modifications of N-glycans such as galactosylation, sialylation, and human natural killer-1 modification[27]. In another study, loss of two GlcNAc branches (β1,2- and β1,6-linked GlcNAc residues on α1,6-Man) in N-glycans by knocking out GnT-II led to hyper-N-acetyllactosamine extension of the remaining GlcNAc branch for functional compensation in T cells[28]. Furthermore, FUT8-KO fibroblasts showed selective elevation of GnT-III activity[29]. These

findings suggest that cells are equipped with regulatory systems for glycosylation activity to adapt to changes in cellular glycosylation.

Recently, proteolytic release of glycosyltransferase into extracellular space has also been found to regulate cellular activity of glycosyltransferases. Although the presence of cleaved soluble forms of glycosyltransferases in body fluids has been known for a long time[30–33], its physiological significance remains elusive. Recently, signal peptide peptidase-like 3 (SPPL3) was identified as being responsible for proteolytic cleavage of various glycosyltransferases[34–36]. Among them, GnT-V is one of the best characterized SPPL3 substrates. It was demonstrated that the C-terminal region of the transmembrane domain of GnT-V is cleaved by SPPL3, resulting in secretion of the protein into the culture medium[34]. Knocking out SPPL3 significantly reduced secretion of GnT-V, thereby elevating the cellular levels of GnT-V protein and its product glycans[34]. This clearly demonstrated the importance of SPPL3-dependent shedding of GnT-V for cellular GnT-V activity. However, it has not been elucidated how this proteolytic cleavage and secretion of GnT-V by SPPL3 is regulated in cells.

In this study, we sought to unveil a novel regulatory mechanism of the activity of N-glycan branching enzymes in cells. We hypothesized that blockade of certain steps in N-glycan maturation in cells specifically induces an activity change of an enzyme for N-glycan branching through compensation or feedback mechanisms. We found that loss of complex-type N-glycans by knocking out various genes or using a glycosylation inhibitor commonly resulted in a specific increase in GnT-V activity. We also found that such activation of GnT-V is caused by impaired cleavage of GnT-V by SPPL3. Our findings highlight a novel regulatory strategy of cells to maintain glycosyltransferase activities.

## Results

### Loss of complex-type N-glycans specifically increased protein expression and activity of GnT-V.
To explore the impact of altered N-glycosylation status on the activities of glycosyltransferases for N-glycan branching, we first treated human embryonic kidney (HEK) 293 cells with a mannosidase inhibitor—kifunensine—to inhibit the conversion of immature oligomannose-type N-glycans to the complex-type N-glycans (Fig. 1a). Conversion of cellular N-glycans by kifunensine treatment was confirmed by the increased reactivity of cellular proteins to mannose-reactive concanavalin A (Con A) lectin[37] and decreased reactivity to leuko-agglutinating phytohemagglutinin (PHA-L4), which recognizes the GnT-V-produced branch[38] (Fig. 1b and Supplementary Fig. 1). To examine the enzyme activities of GnT-III, -IV, -V, and FUT8, lysates of cells treated with or without kifunensine were incubated with their acceptor substrates, GnGnbi-PA (for GnTs) or GnGnbi-Asn-PNS (for FUT8), followed by separation of the products from the unreacted substrates by reverse-phase high-performance liquid chromatography (HPLC) (Fig. 1c), as established previously[39,40]. The retention times of the products of the enzyme reactions were confirmed by control reactions using purified recombinant enzymes (Supplementary Fig. 2a, b). We quantified the activities of these enzymes in the cell lysates by calculating the peak areas of the products, and found that the activities of GnT-III, -IV, and FUT8 in kifunensine-treated cells were almost the same as in untreated cells (Fig. 1c, d). In contrast, GnT-V activity in kifunensine-treated cells was approximately 2.5-times higher than that in untreated cells (Fig. 1c, d), suggesting that GnT-V activity is selectively upregulated by blocking N-glycan maturation. We next examined the expression levels of endogenous GnT-V

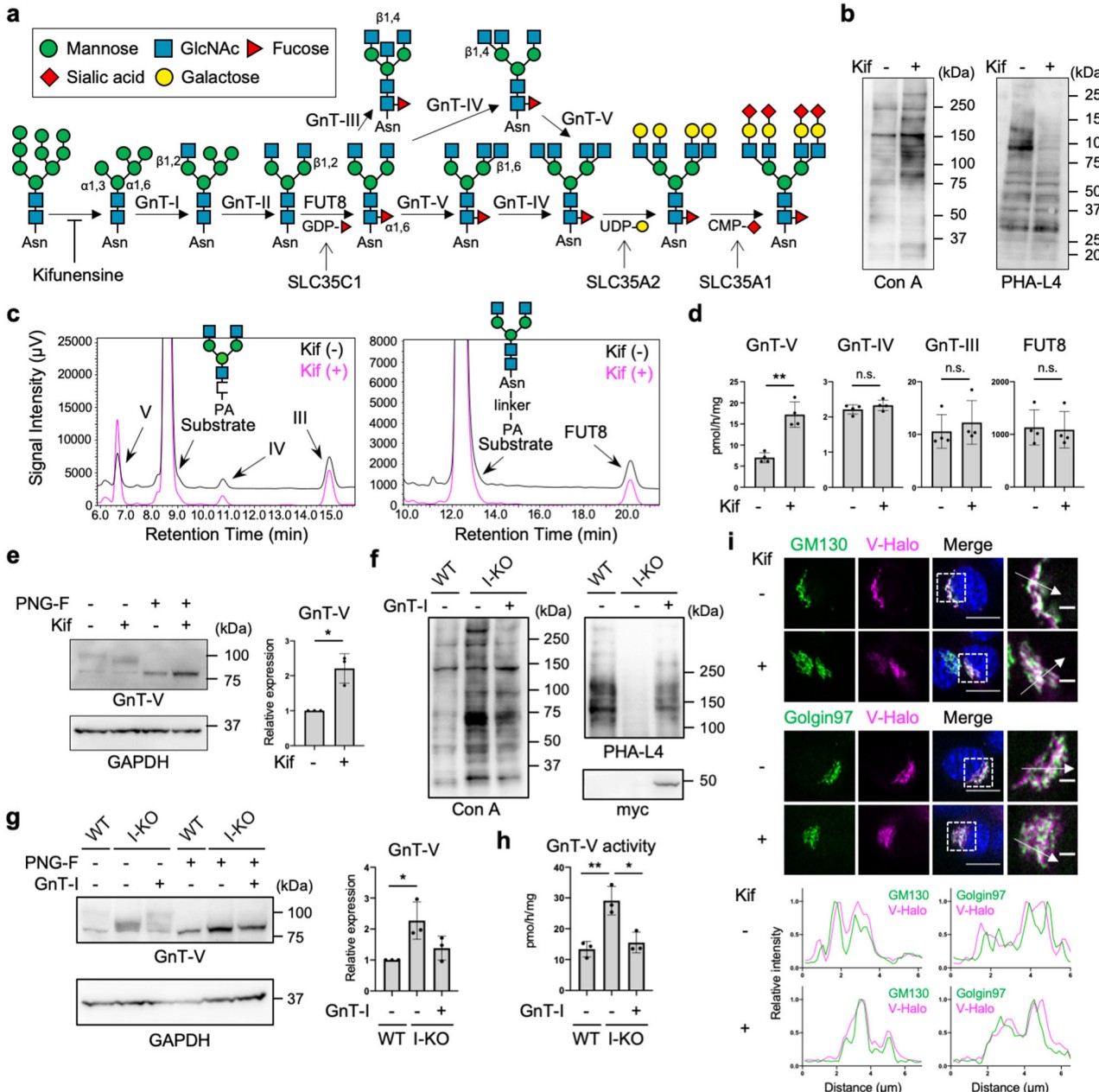

**Fig. 1 Increased protein level and activity of GnT-V in complex-type N-glycan-deficient cells. a** Scheme of biosynthesis pathway of N-glycans. Glycan symbols are according to the standard symbol nomenclature for glycans[93]. Asn: asparagine. **b** Lectin blotting of proteins from HEK293 cells treated with or without kifunensine (kif). **c** In vitro enzyme assays of GnTs (left) and FUT8 (right) in HEK293 cells treated with or without kifunensine. For GnTs, GnGnbi-PA was used as an acceptor substrate. For FUT8, GnGnbi-Asn-PNS was used as an acceptor substrate. The product peak for each enzyme reaction was identified from control reactions using each purified recombinant enzyme, as shown in Supplementary Fig. 1. **d** Quantification of the activities of GnTs and FUT8 as measured in (**c**). Error bars represent SD ($n = 4$). Statistical analysis was by unpaired Student's t-test. **e** Proteins from HEK293 cells treated with or without kifunensine were blotted for GnT-V and GAPDH. Before SDS-PAGE, cell lysates were treated with or without PNGase F (PNG-F). The graph shows quantification of the GnT-V signals in western blots. Error bars represent SD ($n = 3$). Statistical analysis was by Welch's t-test. **f** Proteins from wild-type (WT), GnT-I-deficient (I-KO), and GnT-I (myc-tagged)-rescued HEK293 cells were blotted with Con A, PHA-L4, or anti-myc antibody. **g** Proteins from WT, GnT-I-deficient (I-KO), and GnT-I-rescued HEK293 cells were treated with or without PNGase F and blotted for GnT-V and GAPDH. The graph shows quantification of the GnT-V signals in PNGase F-treated blots. Error bars represent SD ($n = 3$). Statistical analysis was by one-way analysis of variance (ANOVA) with post-hoc Tukey test. **h** The cellular activity of GnT-V in WT, GnT-I-deficient (I-KO), and GnT-I-rescued HEK293 cells. Error bars represent SD ($n = 3$). Statistical analysis was by one-way ANOVA with post-hoc Tukey test. **i** Localization of Halo-tagged GnT-V (V-Halo) visualized in B16 cells treated with or without kifunensine. (Upper) GM130 and Golgin97 were stained as cis- and trans-Golgi markers, respectively. (Lower) Line plots of relative fluorescence intensity are shown. Arrows indicate the plotted regions. Scale bars: 10 or 2 µm. Staining with Calnexin, an ER marker, was shown in Supplementary Fig. 5. *$p < 0.05$; **$p < 0.005$; n.s. not significant.

protein by western blotting in which cell lysates were treated with *N*-glycosidase F (PNGase F) to see the immunoreactive band of GnT-V more clearly by removing *N*-glycans. We found that the level of GnT-V protein in cells was also increased by kifunensine treatment (Fig. 1e), indicating that the loss of complex-type *N*-glycans upregulated GnT-V activity by enhancing the amount of protein in cells.

To examine whether these phenomena were also observable in other cell types, we conducted the same experiments in Neuro-2A cells (a mouse neuroblastoma cell line). The activity and the level of GnT-V protein were both increased by kifunensine treatment (Supplementary Fig. 3a, b). To further examine whether blockade of biosynthesis of complex-type *N*-glycans generally upregulates GnT-V activity in cells, we examined the cellular GnT-V activity in GnT-I-deficient HEK293S cells, from which complex-type *N*-glycans are completely lost[41] (Fig. 1a, f). Both the activity and the level of GnT-V protein were increased in GnT-I-KO cells compared with wild-type (WT) HEK293 cells, to similar levels as in kifunensine-treated HEK293 cells (Fig. 1g, h). Transfection of a GnT-I-myc-tag construct into GnT-I-KO cells cancelled the increased activity and the level of GnT-V protein (Fig. 1g, h), confirming the requirement of GnT-I for regulating the level of GnT-V protein in cells. To further confirm the effects of GnT-I-KO on the level of GnT-V protein, we performed the same experiments in the GnT-I-deficient Chinese hamster ovary (CHO) cell line Lec1[42,43] (Supplementary Fig. 3c). As expected, both activity and the level of GnT-V protein were increased in Lec1 cells compared with WT CHO cells, which was reversed by GnT-I overexpression (Supplementary Fig. 3d, e).

To investigate whether *N*-glycan maturation states also affect GnT-V localization, we compared its localization between control and kifunensine-treated cells. For this purpose, we stably expressed human GnT-V fused with a Halo7 tag (GnT-V-Halo) in B16 cells (a murine melanoma cell line), and we confirmed by in vitro enzyme assays that GnT-V-Halo retained enzyme activity (Supplementary Fig. 4a, b), excluding that addition of the Halo-tag inhibited GnT-V functions. We used B16 cells for these experiments, because HEK293 cells were easily detached during staining. Co-staining with *cis*- and *trans*-Golgi markers GM130 and Golgin97 indicated that GnT-V-Halo signals were well overlapped with both GM130 and Golgin97 signals, particularly with GM130, in cells with or without kifunensine treatment (Fig. 1i). On the other hand, co-staining with an ER marker Calnexin showed that the pattern of GnT-V-Halo signals was clearly different from that of Calnexin with or without kifunensine treatment (Supplementary Fig. 5), suggesting that kifunensine-treatment hardly altered the localization of GnT-V-Halo. Taken together, these results demonstrate that loss of complex-type *N*-glycans specifically increased the protein level and activity of GnT-V without affecting its subcellular localization.

**Secretion of GnT-V by SPPL3-dependent proteolysis is inhibited by blocking N-glycan maturation**. Next, we attempted to elucidate the mechanism of how the level of GnT-V protein was elevated. We first examined the mRNA levels of GnT-V in GnT-I-deficient HEK293S cells. The levels of GnT-V mRNA (encoded by the *MGAT5* gene) were found to be comparable between WT and GnT-I-deficient cells (Fig. 2a), indicating that the level of GnT-V protein is post-transcriptionally upregulated in GnT-I-KO cells. Considering that GnT-I is a Golgi-resident glycosyltransferase and functions in the post-ER organelles, it is less likely that loss of GnT-I affected the general translational regulation machinery. Therefore, we hypothesized that the level of GnT-V protein is post-translationally upregulated by loss of complex-

type *N*-glycans. To further investigate the mechanism, we next focused on degradation of GnT-V protein. HEK293 cells were treated with MG132 (a proteasomal inhibitor) or chloroquine (a lysosomal inhibitor) for 24 h, and the levels of GnT-V protein were examined. While β-catenin and p62, which are subjected to degradation in proteasomes and lysosomes, respectively[44,45], were accumulated in cells treated with these compounds, neither the level of GnT-V protein nor activity in cells was increased by treatment with either compound (Fig. 2b, c). This suggests that degradation of GnT-V protein has at most a minor role in regulation of the level of GnT-V protein. Instead, it was recently reported that GnT-V undergoes proteolytic cleavage by SPPL3 and is subsequently shed into the extracellular space[34]. Thus, we investigated whether the secretion of GnT-V is inhibited in GnT-I-deficient cells. We found that the level of secreted GnT-V in the culture medium of GnT-I-deficient cells was greatly decreased compared with WT HEK293 cells (Fig. 2d), strongly suggesting that shedding of GnT-V is inhibited by the loss of *N*-glycan maturation. We next established an SPPL3-KO cell line; the *Sppl3* gene was knocked out from B16 cells using the CRISPR-Cas9 system. We used B16 cells here, because endogenous GnT-V proteins is more easily detected in B16 than in HEK293 cells. The genotype of the SPPL3-KO clone was confirmed by PCR and genome sequencing (Supplementary Fig. 6a, b). Consistent with previous study[34], knocking out SPPL3 resulted in both abolition of GnT-V secretion and concomitant accumulation of GnT-V protein in cells (Fig. 2e, top row of the left-hand panel), confirming that SPPL3 is the dominant enzyme for GnT-V shedding. Kifunensine treatment of WT cells, which resulted in increased Con A signals and decreased PHA-L4 signals (Fig. 2e, middle and right-hand panels), again significantly increased the level of GnT-V protein and activity in cells (Fig. 2e left-hand panel, first and second lanes, and Fig. 2f), while it decreased the levels of secreted GnT-V (Fig. 2e left-hand panel, fifth and sixth lanes). In sharp contrast, in SPPL3-KO cells, the levels of GnT-V protein (Fig. 2e left-hand panel, third and fourth lanes) and activity (Fig. 2f) in cells were no longer increased by kifunensine treatment. This demonstrates that blockade of *N*-glycan maturation inhibits SPPL3-dependent shedding of GnT-V, leading to the increased levels of GnT-V protein and activity in cells.

**Impact of loss of complex-type N-glycans on the expression, localization, and activity of SPPL3**. We next investigated the mechanism of how cellular *N*-glycan states regulate SPPL3-dependent shedding of GnT-V. We first observed that the level of *SPPL3* mRNA in GnT-I-deficient HEK293S cells was comparable to that in WT or rescued cells (Fig. 3a), excluding that the decreased GnT-V shedding was caused by downregulation of *SPPL3* gene expression. On the basis of these findings, we reasoned that *N*-glycans attached to either SPPL3, GnT-V, or their regulator proteins, affect the broad or GnT-V-specific cleavage activity of SPPL3.

It was previously reported that SPPL3 is not *N*-glycosylated despite the presence of potential *N*-glycosylation sites[46]. To confirm this, *C*-terminally 3 × human influenza hemagglutinin (3HA)-tagged human SPPL3 was expressed in HEK293 cells and detected by western blotting after treatment with or without PNGase F. SPPL3-3HA was detected as a single band in the untreated sample, and the band was not shifted by PNGase F treatment, strongly suggesting that no *N*-glycans are attached to SPPL3 (Fig. 3b). We next investigated whether an *N*-glycosylated protein other than SPPL3 itself affects expression or localization of SPPL3 in an *N*-glycan structure-dependent manner. Recently, it was reported that expression of SPPL3 protein is regulated by the glycoprotein SPPL2c[47], but we found that the levels of SPPL3-

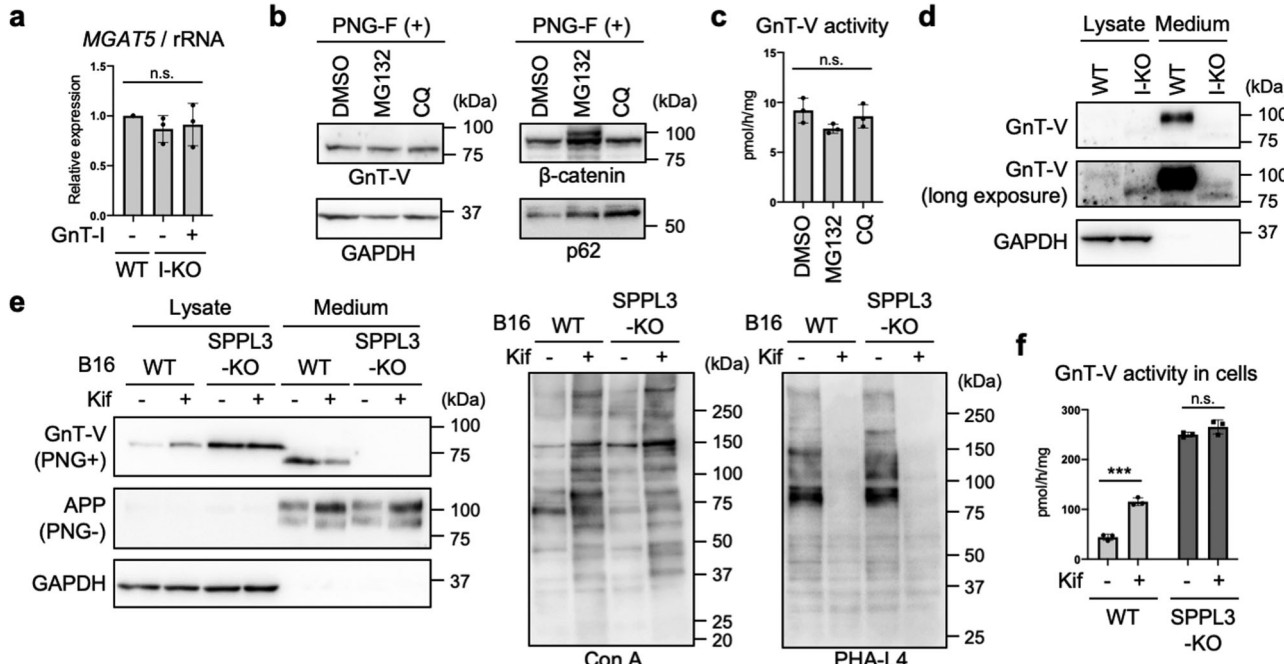

**Fig. 2 Inhibition of SPPL3-dependent secretion of GnT-V in complex-type N-glycan-deficient cells. a** Relative mRNA expression of *MGAT5* (encoding GnT-V) in WT, GnT-I-deficient (I-KO), and GnT-I-rescued HEK293 cells. The levels of *MGAT5* mRNA were normalized by those of rRNA. Error bars represent SD ($n = 3$). Statistical analysis was by one-way ANOVA with *post-hoc* Tukey test. **b** Proteins from HEK293 cells treated with dimethylsulfoxide (DMSO), MG132, or chloroquine (CQ) were blotted for GnT-V, β-catenin, p62, or GAPDH. The samples were treated with PNGase F (PNG-F) before SDS-PAGE. **c** The activity of GnT-V in DMSO-, MG132-, or CQ-treated HEK293 cells. Error bars represent SD ($n = 3$). Statistical analysis was by one-way ANOVA with *post-hoc* Dunnett test. **d** Proteins in the lysates of WT and GnT-I-deficient cells and in the culture media of these cells were blotted for GnT-V or GAPDH. **e** WT and SPPL3-deficient B16 cells were treated with or without kifunensine (kif). Proteins in the cell lysates and the culture media of these cells were blotted for GnT-V, APP, or GAPDH. The samples were treated with or without PNGase F before SDS-PAGE. APP was blotted as a positive control of secreted protein[94]. The proteins in the cell lysates were also blotted with Con A or PHA-L4. **f** The cellular activity of GnT-V in WT and SPPL3-KO B16 cells treated with or without kifunensine. Error bars represent SD ($n = 3$). Statistical analysis was by two-way ANOVA with *post-hoc* Sidak test. ***$p < 0.0005$; n.s. not significant.

3HA protein were not decreased in GnT-I-deficient HEK293S cells (Fig. 3c). Although translational regulation of *SPPL3* mRNA through 5'- or 3'- untranslated regions was not examined in our expression system, it is less likely that knocking out the Golgi enzyme GnT-I directly affects translation of SPPL3 in the ER, because knocking out GnT-I likely affects biological processes within the Golgi or in post-Golgi compartments. Moreover, we observed that the localization of SPPL3-3HA was not altered by kifunensine treatment, the protein being mainly localized to the ER and the Golgi (as also previously reported[48]) regardless of the treatment (Fig. 3d), suggesting that SPPL3 encounters GnT-V in kifunensine-treated cells. As expected, co-localization of SPPL3-3HA and GnT-V-Halo was observed in both kifunensine-treated and -untreated cells (Fig. 3e). These results demonstrate that altered *N*-glycan structures had a negligible effect on the expression and localization of SPPL3.

Next, we investigated whether the activity of SPPL3 was affected by cellular *N*-glycan states. Because SPPL3 has no *N*-glycan, we hypothesized that *N*-glycan structures on GnT-V affect its proteolytic cleavage by SPPL3. To test this hypothesis, we mutated the consensus *N*-glycosylation sites on GnT-V and examined the secretion levels of the mutants. GnT-V has six *N*-glycosylation sites (Fig. 3f), and we deleted these sites by substituting Asn residues to Ser. These mutants exhibited GnT-V activity similarly to that of the WT (Supplementary Fig. 7), suggesting that they mostly maintained proper folding. N110-118S and N334S mutants were secreted at similar levels to WT GnT-V, whereas secretion of N433S and N447S mutants was decreased to approximately 80% of that of the WT (Fig. 3g) (note

that N110-N118S is a mutant in which residues N110, N115, and N118 are all mutated to serine). Together, these results suggest that specific *N*-glycans on GnT-V regulate its shedding by SPPL3.

**Terminal modifications but not branching of N-glycans regulate GnT-V shedding.** Kifunensine treatment or GnT-I KO resulted in loss of all complex-type *N*-glycans, and therefore we next examined the detailed *N*-glycan structures required for the SPPL3-dependent secretion of GnT-V. We first focused on the terminal structures, including fucose, galactose, and sialic acid (Fig. 1a). To investigate the role of fucosylation, we examined the activity of GnT-V in fucosylation-deficient cells. For this purpose, *SLC35C1*, which encodes a GDP-fucose transporter[49], was knocked out from HEK293 cells using the CRISPR-Cas9 system. Genotyping PCR (Supplementary Fig. 8a) and blotting with *Aleuria aurantia* lectin (AAL) (Fig. 4a and Supplementary Fig. 1), which specifically binds to fucosylated glycans[50], demonstrated the loss of fucosylation in SLC35C1-KO cells. We examined cellular GnT-V activity and the secretion of GnT-V, and found that GnT-V activity and the secretion of GnT-V in SLC35C1-KO cells were comparable to those in WT cells (Fig. 4b, c), indicating that fucosylation has no effect on GnT-V shedding. We next examined a previously established galactosylation-defective HEK293 cell line lacking *SLC35A2*, which encodes a UDP-galactose transporter[51,52]. To validate the glycan changes in SLC35A2-KO cells, we investigated the reactivity of the specific lectins, *Griffonia simplicifolia* lectin II (GSL-II) and *Helix pomation* lectin (HPA), which bind to terminal GlcNAc and *N*-

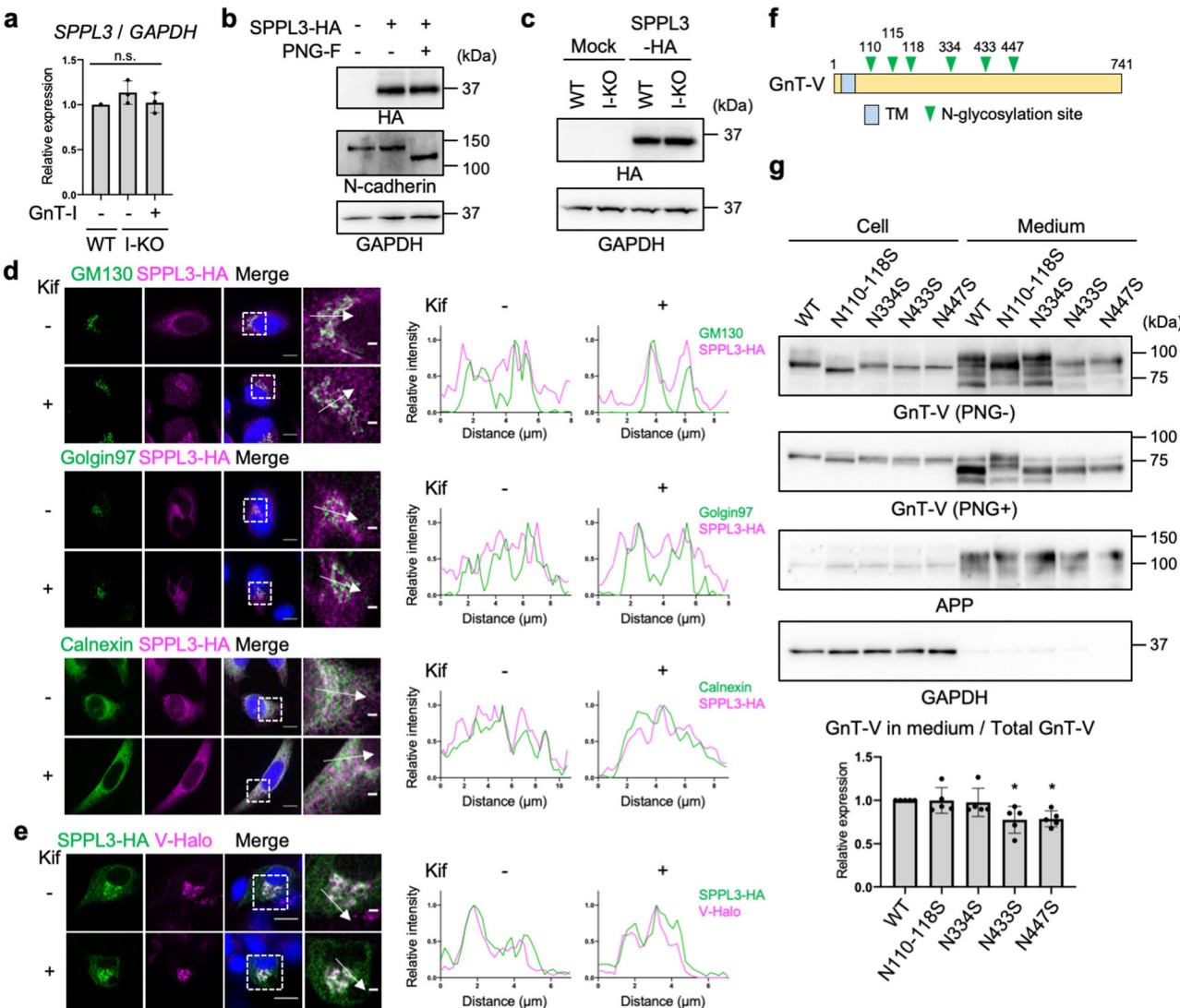

**Fig. 3 Impact of loss of complex-type N-glycans on the expression, localization, and enzyme activity of SPPL3. a** Relative mRNA expression of *SPPL3* in WT, GnT-I-deficient (I-KO), and GnT-I-rescued HEK293 cells. The levels of *SPPL3* mRNA were normalized by those of *GAPDH*. Error bars represent SD ($n = 3$). Statistical analysis was by one-way ANOVA with *post-hoc* Tukey test. **b** SPPL3-3HA was expressed in HEK293 cells, and the lysates were treated with or without PNGase F (PNG-F). The proteins in the lysates were blotted for HA, N-cadherin, or GAPDH. **c** SPPL3-3HA was expressed in WT or GnT-I-deficient cells and the cell lysates were blotted for HA or GAPDH. **d** (Left) HeLa cells expressing SPPL3-3HA by pTK plasmid were treated with or without kifunensine and immunostained for HA and GM130, Golgin97, or Calnexin. (Right) Line plots of relative fluorescence intensity are shown. Arrows indicate the plotted regions. Scale bars: 10 or 2 μm. **e** (Left) B16 cells stably expressing GnT-V-Halo was transfected with SPPL3-3HA and were treated with or without kifunensine. SPPL3-3HA was immunostained with anti-HA antibody and GnT-V-Halo was visualized by TMR. (Right) Line plots of relative fluorescence intensity are shown. Arrows indicate the plotted regions. Scale bars: 10 or 2 μm. **f** Domain structure of human GnT-V. TM: Transmembrane domain. **g** GnT-V WT or its *N*-glycosylation site mutants were expressed in HEK293 cells. Proteins in the cell lysates and culture media of these cells were blotted for GnT-V, APP, or GAPDH. The samples were treated with or without PNGase F before SDS-PAGE. The graph shows quantification of the signal ratio of secreted GnT-V to total (cell + medium) GnT-V in PNGase F-treated blot. APP was blotted as a positive control of secreted protein. Error bars represent SD ($n = 5$). Statistical analysis was by one-way ANOVA with *post-hoc* Dunnett test. *$p < 0.05$; n.s. not significant.

acetylgalactosamine (GalNAc), respectively (Supplementary Fig. 1)[53,54]. Flow cytometry of SLC35A2-KO cells with GSL-II and HPA confirmed the loss of galactosylation, and transfection of human *SLC35A2* gene restored the glycan profiles (Supplementary Fig. 8b). Blotting with galactose-specific *Ricinus communis* agglutinin I (RCA-I)[55] also evidenced the loss of galactosylation from SLC35A2-KO cells (Fig. 4d, left-hand panel). We found that the protein level and activity of GnT-V were both higher in these cells than in WT cells (Fig. 4d middle and right-hand panels, and Fig. 4e), indicating that loss of galactosylation inhibited GnT-V shedding. To investigate the role of sialylation, we generated a sialylation-defective cell line lacking *SLC35A1*,

which encodes a CMP-sialic acid transporter[56,57], using the CRISPR-Cas9 system. Genetic deletion was validated by PCR (Supplementary Fig. 8c). Flow cytometry with *Sambucus nigra* lectin (SNA) and peanut agglutinin (PNA), which preferentially bind to α2,6-linked sialic acid and to *O*-GalNAc glycans with terminally exposed galactose, respectively (Supplementary Fig. 1)[58,59], showed the loss of sialylation in the SLC35A1-KO cell line (Supplementary Fig. 8d). The increased signals from SLC35A1-KO cells in RCA-I blotting also confirmed the loss of sialylation (Fig. 4f left-hand panel). Similar to SLC35A2-KO cells, both the activity and level of GnT-V protein were higher in SLC35A1-KO cells than in WT cells (Fig. 4f middle and right-

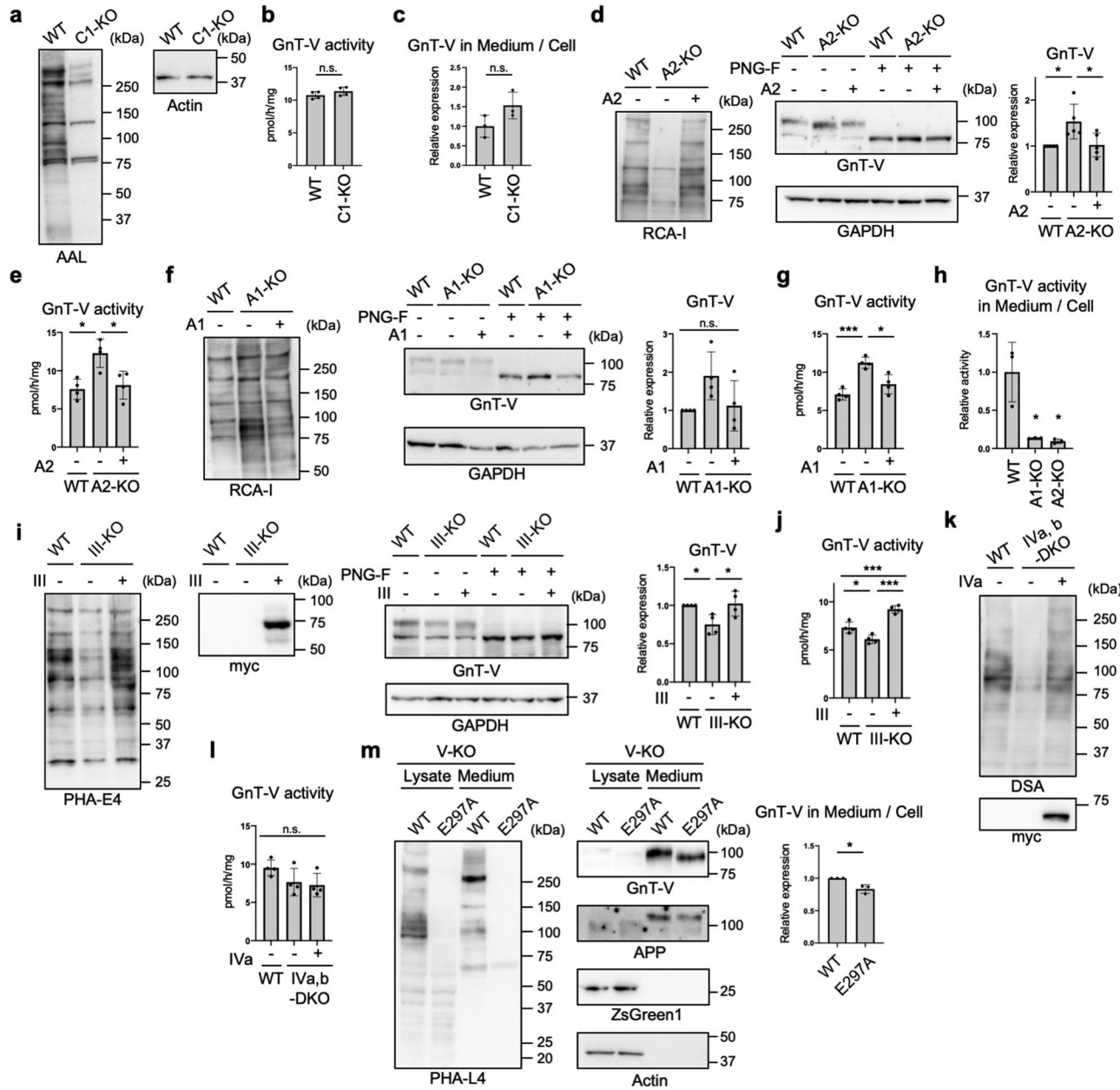

hand panels, and Fig. 4g), indicating that loss of sialylation inhibited GnT-V shedding. Transfection of the relevant genes into each KO cell line restored the altered levels of GnT-V protein and activity (Fig. 4d–g), ruling out off-target effects. Furthermore, GnT-V activity in culture medium was robustly decreased in SLC35A1-KO and SLC35A2-KO cells than in WT cells (Fig. 4h). Because sialic acids are mostly attached to galactose residues and they were depleted in both SLC35A1- and SLC35A2-KO cells, these results strongly suggest that loss of sialylation suppressed GnT-V shedding.

We also examined whether the number of GlcNAc branches in N-glycans affects the SPPL3-dependent secretion of GnT-V. To this end, the responsible enzymes—GnT-III (MGAT3) and -IV (MGAT4A and B)—were knocked out from HEK293 cells, respectively, using the CRISPR-Cas9 system. The recently generated GnT-V (MGAT5)-KO HEK293 cell line[60] was also used. The genotype and loss of enzyme activity of the GnT-III-KO cell line was validated by PCR (Supplementary Fig. 9a), in vitro enzyme assays (Supplementary Fig. 9b), and lectin

blotting (Fig. 4i, left-hand panel). In GnT-III-KO cells, the level of GnT-V protein and activity were not increased compared with that in WT cells (Fig. 4i, j), indicating that bisecting GlcNAc produced by GnT-III has only minor impacts on secretion and cellular expression of GnT-V. Considering GnT-IV, we knocked out both the GnT-IVa and -IVb isoenzymes[61] from HEK293 cells (Supplementary Fig. 9c), and loss of the β1,4-GlcNAc branch from KO cells was confirmed by lectin blotting and flow cytometry using Datura stramonium agglutinin (DSA)[62,63] (Fig. 4k and Supplementary Fig. 9d). In GnT-IVa and -IVb double-KO (DKO) cells, the activity of GnT-V was not increased compared with WT cells (Fig. 4l), suggesting that the β1,4-GlcNAc branch is not involved in GnT-V secretion. To next investigate the effects of the β1,6-GlcNAc branch, which is produced by GnT-V itself, on its secretion, we re-expressed WT GnT-V and a catalytically inactive form of GnT-V (E297A) in GnT-V-KO cells and examined the secretion. We and others recently reported crystal structures of GnT-V and revealed that mutation of the catalytic residue E297 resulted in complete loss of

**Fig. 4 Requirement of glycan terminal modifications for efficient GnT-V shedding. a** Proteins from WT and SLC35C1-KO (C1-KO) HEK293 cells were blotted with AAL or anti-actin antibody. **b** The activity of GnT-V in WT and SLC35C1-KO HEK293 cells. Error bars represent SD ($n = 4$). Statistical analysis was by unpaired Student's $t$-test. **c** $C$-terminally 3 × FLAG-tagged GnT-V WT was expressed in WT and SLC35C1-KO HEK293 cells, and secretion of GnT-V in these cells were assessed by quantify the signal ratio of secreted GnT-V (in medium) to cellular GnT-V. Error bars represent SD ($n = 3$). Statistical analysis was by unpaired Student's $t$-test. **d** Proteins from WT, SLC35A2-KO (A2-KO), and SLC35A2-rescued HEK293 cells treated with or without PNGase F were blotted with RCA-I, and anti-GnT-V and anti-GAPDH antibodies. The graph shows quantification of the GnT-V signals in PNGase F-treated blots. Error bars represent SD ($n = 5$). Statistical analysis was by one-way ANOVA with *post-hoc* Tukey test. **e** The cellular activity of GnT-V. Error bars represent SD ($n = 4$). Statistical analysis was by one-way ANOVA with *post-hoc* Tukey test. **f** Proteins from WT, SLC35A1-KO (A1-KO), and SLC35A1-rescued HEK293 cells treated with or without PNGase F were blotted with RCA-I, and anti-GnT-V and anti-GAPDH antibodies. The graph shows quantification of the GnT-V signals in PNGase F-treated blots. Error bars represent SD ($n = 4$). Statistical analysis was by one-way ANOVA with *post-hoc* Tukey test. **g** The cellular activity of GnT-V. Error bars represent SD ($n = 4$). Statistical analysis was by one-way ANOVA with *post-hoc* Tukey test. **h** Cellular and secreted GnT-V activity from WT, A1-KO and A2-KO HEK293 cells were examined in vitro. The graph shows the relative activity of the ratio of secreted GnT-V (in medium) to cellular GnT-V. Error bars represent SD ($n = 3$). Statistical analysis was by one-way ANOVA with *post-hoc* Dunnett test. **i** Proteins from WT, GnT-III-KO (III-KO), and GnT-III (myc-tagged)-rescued HEK293 cells treated with or without PNGase F were blotted with PHA-E4, and anti-myc, anti-GnT-V, and anti-GAPDH antibodies. The graph shows quantification of the GnT-V signals in PNGase F-treated blots. Error bars represent SD ($n = 4$). Statistical analysis was by one-way ANOVA with *post-hoc* Tukey test. **j** The cellular activity of GnT-V. Error bars represent SD ($n = 4$). Statistical analysis was by one-way ANOVA with *post-hoc* Tukey test. **k** Proteins from WT, GnT-IVa and IVb-DKO (IVa,b-DKO), and GnT-IVa (myc-tagged)-rescued HEK293 cells were blotted with DSA and anti-myc antibody. **l** The cellular activity of GnT-V. Error bars represent SD ($n = 4$). Statistical analysis was by one-way ANOVA with *post-hoc* Tukey test. **m** GnT-V WT or its catalytically inactive E297A mutant was expressed in GnT-V-KO (V-KO) HEK293 cells. Proteins in the cell lysates and the culture media of these cells were blotted with PHA-L4, and anti-GnT-V, anti-APP, anti-ZsGreen1, or anti-actin antibodies. The graph shows quantification of the signal ratio of secreted GnT-V to cellular GnT-V. APP was blotted as a positive control of secreted protein. Error bars represent SD ($n = 3$). Statistical analysis was by Welch's $t$-test. $*p < 0.05$; $***p < 0.0005$; n.s. not significant.

activity[64,65]. Consistent with this, expression of the E297A mutant did not restore PHA-L4 reactivity in GnT-V-KO cells (Fig. 4m left-hand panel). We observed that secretion of E297A mutant was only slightly decreased compared with the WT enzyme (Fig. 4m). Similar levels of ZsGreen1 (whose cDNA was inserted downstream of *MGAT5*-IRES2 sequence) in WT and E297A-expressing cells confirmed the similar transfection efficiencies of the WT- and E297A mutant-encoding plasmids (Fig. 4m). These data indicated that the β1,6-GlcNAc branch has only minor effects on GnT-V secretion. Collectively, our findings suggest that terminal galactosylation and sialylation, but not GlcNAc branching, enhance GnT-V shedding.

**Enhanced GnT-V activity increases N-glycan branching in cells.** To investigate whether the enhanced cellular activity of GnT-V resulted in increased GlcNAc branching of *N*-glycans in cells, we performed *N*-glycomic analysis. To this end, we compared *N*-glycan structures between WT cells and SLC35A2-KO cells in which cellular GnT-V activity was upregulated (Fig. 4d). *N*-Glycans were released from these cells, labeled with aminoxyTMT6 reagent, desialylated, and analyzed by liquid-chromatography–electrospray ionization-mass spectrometry (LC-ESI-MS). Glycomic data demonstrated that the overall *N*-glycan profile of SLC35A2-KO (A2-KO) cells was totally different from that in WT and rescued (A2KO + SLC35A2) cells, which mainly resulted from the loss of galactosylation in the SLC35A2-KO cells (Supplementary Fig. 10a). To more directly compare relative amounts of each number of GlcNAc branches, we simplified the glycan structures by removing galactose using β-galactosidase (Fig. 5a). Relative amounts of the complex-type glycans classified by the number of *N*-acetylhexosamine (HexNAc) units in antennas clearly showed that the less branched structures (having two or three HexNAcs) were decreased in KO cells, whereas the more highly branched structures (having four or five HexNAcs) were increased in KO cells (Figs. 5b, S10b, and Supplementary Data 2). By comparing with enzymatically prepared standard glycans, we further identified the GnT-V products among the glycans with three and four HexNAc residues (Supplementary Fig. 10c), and found that the levels of GnT-V products having four or five HexNAcs were drastically increased in KO cells (Fig. 5c). This demonstrates that the GnT-V activity enhanced in

cells by blocking *N*-glycan maturation resulted in an increase in its product glycans.

**Glycan-dependent regulation of SPPL3-mediated shedding is substrate-selective.** Finally, we investigated whether secretion of SPPL3 substrates other than GnT-V is also regulated by cellular *N*-glycan structures. Previously, CANT1 and B4GALT1, both of which are type-II transmembrane proteins involved in glycosylation[66–68], were reported to be secreted in an SPPL3-dependent manner[34,35]. Consistent with this, we observed that secretion of these enzymes was almost completely inhibited in SPPL3-KO HEK293 cells (Fig. 6a) which were generated by CRISPR-Cas9 (Supplementary Fig. 6c). We then investigated whether secretion of these enzymes was also suppressed in GnT-I-deficient cells. In contrast to the severe inhibition of GnT-V shedding in such conditions, secretion of CANT1 was partially inhibited in GnT-I-KO cells, and B4GALT1 secretion was barely affected (Fig. 6b). These results suggest that regulation of SPPL3-mediated shedding is substrate-dependent, and that shedding of multiple glycan-related enzymes, including GnT-V, are selectively regulated by cellular *N*-glycan structures.

**Discussion**
Here, we demonstrate that the level of GnT-V protein and activity in cells are regulated by *N*-glycan-dependent shedding by SPPL3. Our glycomic analysis revealed that accumulation of GnT-V protein in cells resulted in a significant increase in the amount of highly branched *N*-glycans, indicating that this shedding-based regulatory mechanism impacts cellular *N*-glycan profiles. Furthermore, because SPPL3 is ubiquitously expressed[69], this regulation of GnT-V is likely the general mechanism, and we indeed found that upregulation of GnT-V activity by inhibiting *N*-glycan maturation was observed in four different cell lines: HEK293 (Figs. 1c–h, 4), Neuro-2A (Supplementary Fig. 3a, b), CHO (Supplementary Fig. 3c–e), and B16 (Fig. 2e, f) cells. Therefore, this regulatory mechanism could contribute to the alteration of cellular *N*-glycan profiles in various physiological and pathological processes.

The cellular activity of GnT-V is regulated by several mechanisms. Considering transcriptional regulation, an

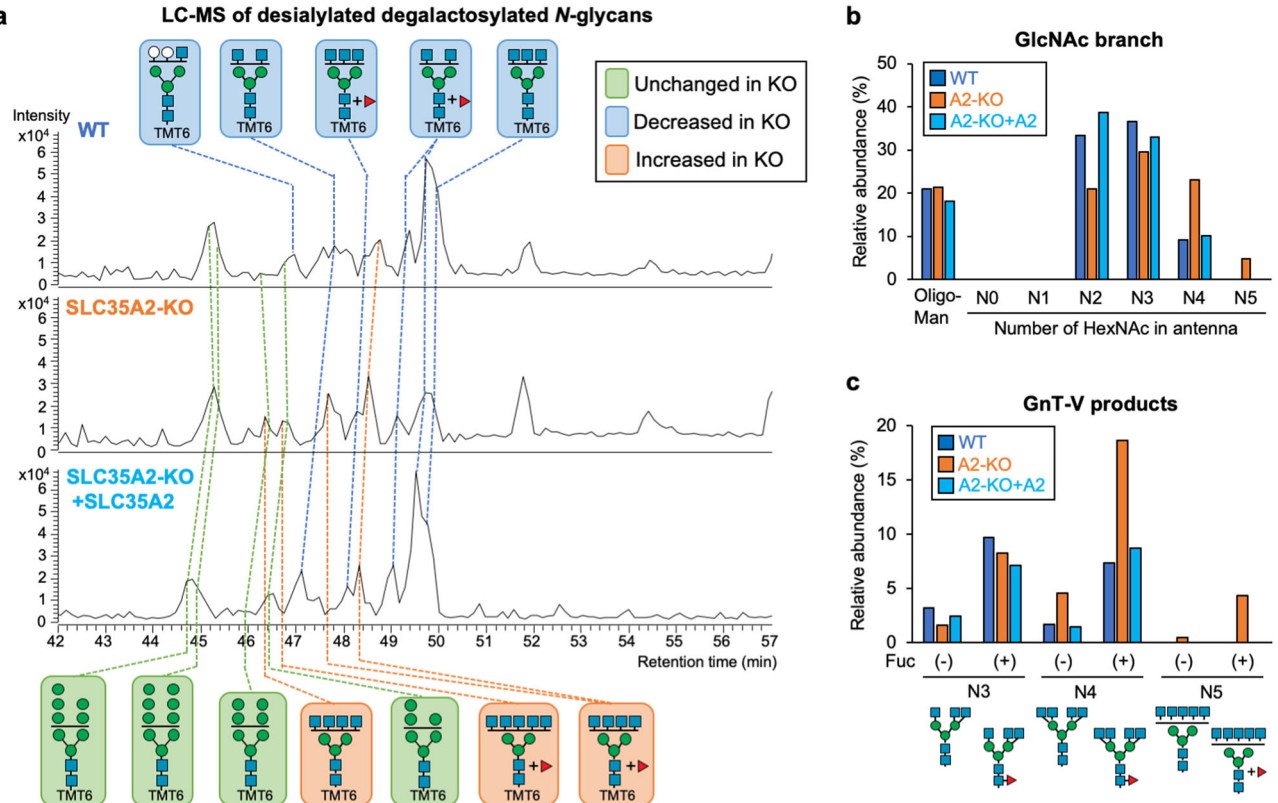

**Fig. 5 N-Glycomics of WT, SLC35A2-KO, and SLC35A2-rescued HEK293 cells. a** Base peak chromatograms (BPCs) from LC-ESI MS analysis of TMT6-labeled desialylated β-galactosidase-treated N-glycans from WT, SLC35A2-KO (A2-KO), and SLC35A2-rescued (A2-KO + A2) HEK293 cells. The deduced structures of the major N-glycans are shown. Orange, glycans that increased in A2-KO cells compared with in WT cells; blue, glycans that decreased in A2-KO cells compared with in WT cells; green, glycans unchanged between WT and A2-KO cells. **b** The signal intensities of oligomannose glycans, and complex-type glycans with 0, 1, 2, 3, 4, and 5 HexNAc residues (HexNAc in chitobiose is excluded) from LC-MS analysis. **c** The signal intensities of GnT-V product glycans with 3, 4, and 5 HexNAc residues (HexNAc in chitobiose is excluded) with or without a Fuc residue from LC-MS analysis.

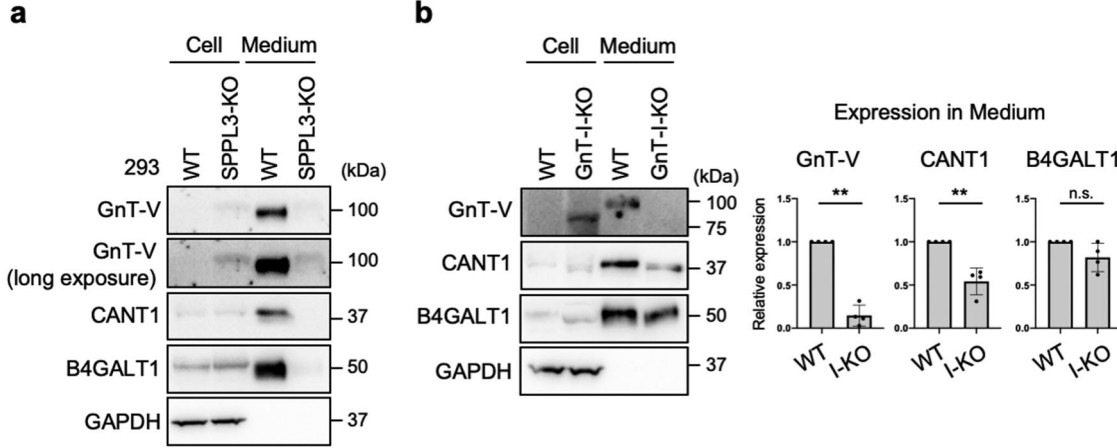

**Fig. 6 Secretion of other SPPL3 substrates from GnT-I-KO cells. a** Proteins in the lysates of WT and SPPL3-deficient HEK293 cells and in the culture medium of these cells were blotted for GnT-V, CANT1, B4GALT1, or GAPDH. **b** Proteins in the lysates of WT and GnT-I-deficient (I-KO) HEK293 cells and in the culture medium of these cells were blotted for GnT-V, CANT1, B4GALT1, or GAPDH. The graphs show quantification of the signal intensities of GnT-V, CANT1, and B4GALT1 secreted from I-KO cells relative to secretion from WT cells. Error bars represent SD ($n = 4$). Statistical analysis was by Welch's $t$-test. **p < 0.005; n.s. not significant.

oncogenic signaling cascade—the Ras–Raf–Ets pathway—drives the *MGAT5* gene[70,71], which suggests involvement of GnT-V in cancer. In contrast, post-translational regulation of GnT-V activity is poorly understood. In general, formation of homo- and hetero-complexes and control of subcellular localization play

important roles in the regulation of glycosyltransferase activity[20,26]. However, homodimerization was reported to be dispensable for the activity of GnT-V[24], and there has been little study of proteins that interact with GnT-V to date. Regarding subcellular localization, although an L188R point mutant of GnT-

V was found to mislocalize to the ER and lose activity in cells[72], it has not been fully clarified how subcellular localization of GnT-V is regulated and impacts its activity. Recent work and our present study demonstrated that loss of SPPL3 greatly upregulated the levels of cellular GnT-V protein and its product glycans[34]. This suggests that shedding of GnT-V mediated by SPPL3 significantly contributes to regulation of GnT-V activity in cells. Furthermore, because secreted soluble GnT-V was reported to have angiogenic activity[33], GnT-V shedding may have a dual role: control of both cellular GnT-V activity and alteration of the extracellular milieu through another function.

Our present work clarified the unexpected link between terminal glycan modifications and the GnT-V-produced branch. The SPPL3-mediated regulation of GnT-V activity could explain why the number of GlcNAc branches in N-glycans was elevated in patients harboring SLC35A2 mutations[73]. Moreover, recent studies have revealed several unexpected inter-regulation mechanisms between different types of glycan. For instance, total cellular glycomics in CHO, Lec1 (GnT-I deficient CHO), and Lec8 (SLC35A2-deficient CHO) cells demonstrated that O-GalNAc glycans, glycosaminoglycans, and glycolipids were unpredictably altered by loss of complex-type N-glycans or galactosylation[74]. More recent work using a KO cell library of glycan-related enzymes also showed that knocking out of N-glycan mannosidase genes resulted in elevation of glyco-sphingolipids and hyaluronan[75]. The accumulating evidence of this crosstalk suggests that biosynthesis of each type of glycan is regulated by other glycosylation pathways. As SPPL3 cleaves glycan-related enzymes that are involved with N-glycans, O-GalNAc glycans, glycosaminoglycans, and glycolipids[34–36], SPPL3-mediated shedding could explain the mechanisms of this crosstalk by regulation of the various glycosylation pathways both directly and indirectly.

SPPL3 is an intramembrane protease categorized as a GXGD aspartyl proteases, of which there are seven family members in humans: Presenilin-1 and −2, SPP, and four SPPLs (SPPL2a, SPPL2b, SPPL2c, and SPPL3)[76]. Recently, transmembrane proteins, including tail-anchored proteins, type-II transmembrane proteins, and multipass transmembrane proteins were found to be substrates of SPP and SPPLs[77], and SPPL3 was found to be the dominant sheddase for glycan-related enzymes[77]. However, it has not been elucidated how SPPL family members recognize their substrates. Our results, in which SPPL3 cleaved GnT-V in a N-glycan structure-dependent manner (Figs. 4 and 6b), raise the possibility that SPPL3 may recognize specific N-glycan structures in its substrates. However, given that SPPL3 has nine trans-membrane domains with small luminal regions, and that the GnT-V N-glycans for SPPL3-dependent secretion (those attached to N433 and N447) are located away from the membrane[64,65], it is unlikely that SPPL3 itself directly recognizes N-glycans. Instead, we speculate that SPPL3 forms a functional complex with a lectin-like protein that recognizes the N-glycan structures of SPPL3 substrates, allowing substrate-selective activity. Recent work showed that SPP family members form hetero-complexes for substrate recognition. For example, SPP has been shown to form a large complex (~500 kDa) with components of the ER-associated degradation (ERAD) machinery, Derlin1 and TRC8[78], which is distinct from a 200 kDa complex for cleavage of signal peptides of nascent proteins[78]. Because Derlin1 is thought to function as a receptor for the unfolded luminal domain of XBP1u[78–81], differential hetero-complex formation may be ben-eficial to distinguish ERAD substrates and signal peptides. SPPL2c was also reported to form a higher order complexes (~500 kDa) distinct from the SPP-complex described above[82]. Although the precise components of this hetero-complex remain unknown, complex formation may be involved in the substrate

selectivity. For further understanding of how SPPL3 selectively acts on its substrates, it will be important to identify proteins that interact with SPPL3.

Our analyses using a series of glycan-deficient cells showed that terminal modifications of N-glycans, including galactosylation and sialylation, efficiently enhance SPPL3 cleavage (Fig. 4). However, these terminal modifications are present in most N-glycoproteins. Moreover, SPPL3-mediated shedding of B4GALT1, which also has N-glycans, occurred almost normally even in GnT-I-KO cells (Fig. 6b). On the basis of these findings, it is likely that SPPL3-mediated cleavage does not solely depend on galac-tosylation and sialylation, but rather it is regulated by multiple factors, such as other parts of glycans, polypeptides, or both. One possibility is that the combination of both a β1,6-branch and a β1,4-branch (produced by GnT-IV) may enhance cleavage by SPPL3 because GnT-V secretion was partially impaired in β1,6-branch-deficient conditions (Fig. 4m). Consistently, a previous site-specific glycomic study of human GnT-V demonstrated that N443 and N447 were preferentially modified with highly bran-ched tri- and tetra-antennary glycans, while N334 was modified with only biantennary glycans[83]. Furthermore, we revealed that the protein level and the cellular activity of GnT-V were slightly decreased in GnT-III-KO cells compared with those in WT cells (Fig. 4i, j), suggesting that loss of bisecting GlcNAc promotes GnT-V shedding. Because the presence of bisecting GlcNAc suppresses the reactions of various glycosyltransferases, including GnT-IV and GnT-V[27], loss of bisecting GlcNAc likely resulted in elevation of the number of GlcNAc branches, which could in turn enhance GnT-V cleavage by SPPL3. It will be intriguing to investigate whether SPPL3-dependent secretion of GnT-V is impaired by double knockout of the β1,6-branch and β1,4-branch.

In summary, we report a novel secretion-based regulatory mechanism of cellular N-glycosylation. Because post-translational cleavage is more rapid than transcriptional regulation, the system has the potential to rapidly alter the N-glycan status in cells. Because SPPL3 has unique specificity toward a wide variety of glycan-related enzymes, it is of great interest to elucidate how its activity is regulated and dysregulated in disease conditions. Fur-ther analyses will be needed to understand the detailed mechanisms of SPPL3-mediated regulation of glycosylation, including how SPPL3 cooperates with other proteins to selectively act on its substrates, and how the activity and specificity of SPPL3 are regulated by various classes of glycans in cells.

## Method
**Antibodies and reagents**. The following antibodies were used in this study: anti-GnT-V monoclonal antibody [mAb; clone 24D11;[84] a gift from Dr. Eiji Miyoshi (Osaka University, Osaka, Japan)]; anti-GAPDH mAb (clone 6C5), anti-Amyloid precursor protein (APP) (clone 22C11), anti-actin mAb (clone AC-40), and anti-myc-tag mAb (clone 4A6) from Millipore (MAB374, MAB348, A4700, and 05-724); anti-β-catenin mAb (clone 14) and anti-GM130 mAb (clone 35) from BD Biosciences (610154 and 610822); rabbit anti-Sequestosome 1 (SQSTM1)/p62, rabbit anti-Golgin97 (clone D8P2K), and rabbit anti-HA-tag mAb (clone C29F4) from Cell Signaling Technology (5114, 13192, and 3724); anti-GnT-V mAb (clone 706824), anti-CANT1 mAb (clone 861206), and goat anti-B4GALT1 from R&D systems (MAB5469, MAB6720, and AF3609); anti-FLAG mAb (clone M2) from Sigma-Aldrich (F1804); anti-ZsGreen1 mAb (clone 2C2) from ORIGENE; rabbit anti-calnexin (ab22595) from Abcam; rat anti-HA (clone 3F10) from Merck (11867423001); horseradish peroxidase (HRP)-conjugated anti-mouse immu-noglobulin (Ig) G and HRP-conjugated anti-rabbit IgG from GE Healthcare (NA931V and NA934V); HRP-conjugated anti-goat IgG from Jackson Immu-noResearch Laboratories (705-035-147); and Alexa488-conjugated anti-mouse IgG, Alexa488-conjugated anti-rabbit IgG, Alexa546-conjugated anti-mouse IgG, and Alexa546-conjugated anti-rat IgG from ThermoFisher Scientific (A21202, A21206, A10036, and A11081).

The following lectins were used in this study: biotinylated Con A, DSA, erythro-agglutinating phytohemagglutinin (PHA-E4), and PHA-L4 from J-Chemical (J203, J105, J111, and J112); biotinylated PNA from J-Oil Mills (300430); biotinylated AAL, biotinylated RCA-I, fluorescein isothiocyanate (FITC)-conjugated GSL-II,

and FITC-conjugated SNA from Vector Laboratories (B-1395, B-1085, FL-1211, and FL-1301); and Alexa488-conjugated HPA from ThermoFisher Scientific (L11271).

The following reagents and kits were used: kifunensine (Cayman, 10009437); Halo-tag TMR ligand (Promega, G8251); MG-132 (Millipore, 474790); chloroquine diphosphate (Wako, 038-17971); PNGase F and cOmplete Mini Protease Inhibitor Cocktail [ethylenediaminetetraacetic acid (EDTA)-free] (Roche, 11365169001 and 11836170001); a VECTASTAIN ABC kit (from Vector Laboratory, PK-4000); a QuickChange Lightning Site-Directed Mutagenesis kit (Agilent Technologies, 210518-5); a Pierce BCA Protein Assay Kit, Lipofectamine 3000, and the SuperScript IV First-Strand Synthesis System (ThermoFisher Scientific, 23227, L3000 and 18091050); Polyethylenimine (PEI) MAX (Polysciences, 24765); anti-DYKDDDDK-tag antibody Magnetic Beads (FLAG beads; Wako, 017-25151); and NEBuilder HiFi DNA Assembly Master Mix (New England Biolabs, E2621).

**HRP-labeling of lectins.** HRP-labeling of DSA, PHA-E4, and PHA-L4 was performed with a Peroxidase Labeling Kit–NH$_2$ (DOJINDO) according to the manufacturer's protocol. Briefly, 200 µg of nonlabelled lectin was diluted in 100 µL of the washing buffer (supplied in the kit), and the samples were applied to the filtration tubes from the kit, followed by centrifugation at 8000 × g for 10 min. Membranes in the filtration tubes were then washed with 100 µL of washing buffer by centrifugation at 8000 × g for 10 min. Peroxidase was mixed in 10 µL of the reaction buffer (supplied in the kit), and the mixture was applied onto the membrane, followed by incubation at 37 °C for 2 h. Washing buffer (100 µl) was added to the mixture, followed by centrifugation at 8000 × g for 10 min. Finally, HRP-labeled lectins were eluted with 200 µL of storage buffer (supplied in the kit) by pipetting at least 10 times onto the membranes.

**Plasmid construction.** Primers used in this study are listed in Supplementary Data 1. cDNAs of human SLC35A1 and SPPL3 were amplified from a cDNA library of HEK293 cells with primer sets #1-#2 and #3-#4, respectively. The amplified sequences were digested with EcoRI/NotI and EcoRI/MluI, respectively, followed by ligation with pcDNA6/myc-His A and pME-3HA, respectively, which had been digested with the same restriction enzyme pairs. To construct pcDNA6/myc-His A-hSPPL3-3HA, a fragment containing human (h) SPPL3 cDNA was amplified using pME-hSPPL3-3HA as the template with primers #3-#5, followed by digestion with EcoRI and NotI. The fragment was inserted into pcDNA6/myc-His A with the same restriction enzyme pair. To construct C-terminal Halo-tagged hGnT-V, the sequences encoding hGnT-V and the Halo-tag were amplified using pcDNA6/myc-His A-hGnT-V (nontag)[85] and pEGFPN1-ITGB1-SG×3-Halo7[86] as the templates with primer sets #6-#7 and #8-#9, respectively. The amplified sequences were subcloned into pcDNA6/myc-His A digested with NotI and XhoI using NEBuilder HiFi DNA Assembly Master Mix. To construct p3×FLAG-hGnT-V, hGnT-V sequence was amplified with primer sets #10-#11 using pSVK3/hGnT-V as a template, followed by digestion with NotI/XbaI. The fragment was inserted into p3×FLAG-CMV14 digested with the same enzyme pair. The plasmids for point mutants of hGnT-V were generated using pcDNA6/myc-His A-hGnT-V (nontag) as the template and a QuickChange Lightning Site-directed Mutagenesis kit, with primers: N110-118S, #12-#13; N334S, #14-#15; N433S, #16-#17; and N447S, #18-#19. To construct pTK-hGnT-V-IRES2-ZsGreen1, the linearized pTK-hGnT-V sequence was obtained by PCR using pTK-hGnT-V[85] as the template and primers #20-#21, and the IRES2-ZsGreen1 sequence was amplified by PCR using pIRES2-ZsGreen1 (Takara) as the template and primers #22-#23. The amplified sequences were ligated using NEBuilder HiFi DNA Assembly Master Mix. pTK-hGnT-V(E297A)-IRES2-ZsGreen1 was generated from pTK-hGnT-V-IRES2-ZsGreen1 by site-directed mutagenesis using the QuickChange Lightning Site-Directed Mutagenesis Kit and primers #24-#25. To construct pTK-hSPPL3, hSPPL3-3HA sequence was amplified by PCR using primers #26-#27. The fragment was inserted into pTK plasmid digested with XhoI and XbaI using NEBuilder HiFi DNA Assembly Master Mix. To construct plasmids for gene editing, pX330-EGFP[87] was digested with BbsI followed by ligation with the annealed primer sets listed: sgRNA for hSLC35A1#1, #28-#29; hSLC35A1#2, #30-#31; hSLC35C1#1, #32-#33; hSLC35C1#3, #34-#35; hMGAT3 (encoding GnT-III) #1, #36-#37; hMGAT3#2, #38-#39; mSppl3#1, #40-#41; mSppl3#2, #42-#43; hSPPL3#1, #44-#45; and hSPPL3#2, #46-#47. The sgRNAs were designed using CRISPRdirect, E-CRISP, and CRISPOR software.

**Cell culture.** HEK293, Neuro-2A, GnT-I-deficient HEK293S[41], HeLa, CHO, and Lec1[43] cells were obtained from the American Type Culture Collection, and COS7 cells and B16 cells were purchased from RIKEN cell bank. Cell line HEK293-SLC35A2-KO[88] was kindly gifted by Dr. Taroh Kinoshita (Osaka University, Osaka, Japan). Cell line HEK293, its derivatives [GnT-III-KO (clone #16), GnT-IVa and IVb-DKO (clone #12), GnT-V-KO (clone #10), SLC35A1-KO (clone #12), SLC35A2-KO, or SLC35C1-KO (clone #4), and SPPL3-KO (clone #1)], GnT-I-deficient HEK293S, Neuro-2A, HeLa, COS7, and B16 and its derivative [SPPL3-KO (clone #1)] were cultured in Dulbecco's modified Eagle's medium (DMEM) supplemented with 10% fetal bovine serum at 37 °C under 5% (v/v) CO$_2$. CHO and Lec1 cells were cultured in alpha-modified Eagle's medium supplemented with 10% fetal bovine serum in the same conditions as described above.

To abrogate complex-type N-glycans in cells, HEK293, Neuro-2A, and B16 cells were treated with kifunensine (5 µM for HEK293 and Neuro-2A, and 10 µM for B16) for 48 h. To inhibit protein degradation pathways, HEK293 cells were treated with 10 µM MG-132 or 50 µM chloroquine for 24 h.

**Establishment of KO cells and stable transfectants.** To generate SLC35A1-KO or SPPL3-KO cell lines, two different pX330-enhanced Green Fluorescent Protein (EGFP)-based plasmids harboring sgRNAs targeting each gene were transfected as described below. Two days after transfection, cells with high expression of EGFP were collected by cell sorting using FACS Melody apparatus (BD Biosciences). To generate SLC35C1-KO or GnT-III-KO cell lines, two different pX330-puro plasmids harboring sgRNAs targeting each gene were transfected as described below. One day after transfection, cells were selected with 3 µg/ml puromycin. The sgRNA sequences used are listed in Supplementary Data 1 (#28 to #47). All KO cell lines were established by cloning by limiting dilution. Knock-out of genes was confirmed by PCR using the following primers (listed in Supplementary Data 1): SLC35A1, #48-#49; SLC35C1, #50-#51; MGAT3 (GnT-III), #52-#53; Sppl3, #54-#55 for B16 cell line (mouse); SPPL3, #56-#57 for HEK293 cell line (human).

To generate cells stably re-expressing SLC35A2 in SLC35A2-KO cells, pcDNA6/myc-His A and pcDNA6/myc-His A-hSLC35A2 were transfected into SLC35A2-KO cells for mock and hSLC35A2 transfectants, respectively. Two days after transfection, cells were selected with 8 µg/ml blasticidin. After 1 week of culture, SLC35A2 transfectants were further selected by sorting in which staining with GSL-II was negative. To generate cells stably expressing hGnT-V-Halo, pcDNA6/myc-HisA-hGnT-V-Halo7 was transfected into B16 cells and selected with 8 µg/ml blasticidin.

**Plasmid transfection.** Cells at approximately 50% confluency grown on 6- or 10 cm dishes were transfected with plasmid using Lipofectamine 3000 reagent according to the manufacturer's protocol.

**Sample preparation.** Cells were washed with phosphate-buffered saline (PBS) twice and collected using cell scrapers, followed by centrifugation at 1400 × g for 3 min. The cell pellets were lysed by sonication in lysis buffer [20 mM Tris (pH 7.4), 150 mM NaCl, 1% Triton X-100] containing protease inhibitor cocktail. Protein concentration was measured using a BCA kit.

For preparation of secreted proteins, culture medium (serum-free DMEM) was collected by centrifugation at 1200 × g for 5 min to remove cell debris. The supernatants were concentrated with Amicon-Ultra filter units (cut-off 10 kDa; Millipore) by centrifugation at 3900 × g for 30 min at 4 °C. The concentrated samples were then mixed with 2.5 volumes of 100% ethanol and 1/30 volume of 5 M NaCl, followed by incubation at –80 °C for 10 min. Proteins were then precipitated by centrifugation at 13,800 × g for 20 min at 4 °C. The pellet was washed with 70% ethanol and centrifuged at 13,800 × g for 5 min at 4 °C. The protein pellet was dissolved by sonication in lysis buffer containing protease inhibitor cocktail. Protein concentrations were measured using a BCA kit.

For PNGase F treatment, samples were denatured in denaturing buffer [20 mM Tris (pH 7.4), 0.5% sodium dodecyl sulfate (SDS), 1% β-mercaptoethanol, 5 mM EDTA] at 95 °C for 5 min, followed by fivefold dilution with Tris-buffered saline (TBS) containing Nonidet P-40 (NP-40; final concentration, 0.5% v/v). Samples (60 µL) were then incubated with 3 µL of water or PNGase F at 37 °C for >2 h. The samples were then mixed with 5× Laemmli SDS sample buffer and incubated at 95 °C for 5 min.

**Measurement of glycosyltransferase activities.** Activities of GnT-III, -IV, and -V and FUT8 in cell lysate were measured as described previously[39,40]. For determination of GnT-III and -IV activities, cell lysates were incubated in 10 µL of GnT-III reaction buffer [125 mM 2-Morpholinoethanesulfonic acid (MES; pH 6.25), 10 mM MnCl$_2$, 200 mM GlcNAc, 0.5% Triton X-100, and 1 mg/ml bovine serum albumin (BSA)] supplemented with 20 mM UDP-GlcNAc and 10 µM fluorescently-labeled biantennary acceptor N-glycan substrate [GnGnbi-PA (PA, 2-aminopyridine)] at 37 °C for 16 h. For determination of GnT-V activity, cell lysates were incubated in 10 µL of GnT-V reaction buffer [125 mM MES (pH 6.25), 10 mM EDTA, 200 mM GlcNAc, 0.5% Triton X-100, and 1 mg/ml BSA] supplemented with 20 mM UDP-GlcNAc and 10 µM acceptor substrate (GnGnbi-PA) at 37 °C for 16 h (or 3 h if GnT-V was transfected). For determination of FUT8 activity, cell lysates were incubated in 10 µL of FUT8 reaction buffer [100 mM MES (pH 7.0), 200 mM GlcNAc, 0.5% Triton X-100, and 1 mg/ml BSA] supplemented with 1 mM GDP-fucose and 10 µM fluorescently-labeled biantennary acceptor N-glycan substrate [GnGnbi-Asn-PNS (PNS, N-(2-(2-pyridylamino)ethyl)succina-midyl)] at 37 °C for 30 min. To identify product peaks from each enzyme in HPLC, reactions using recombinant enzymes were performed as described previously. Recombinant truncated forms of hGnT-III (from Glu63 to Val533), mouse GnT-IVa (from Ser60 to Ser526), hGnT-V (from Thr121 to Leu741), and hFUT8 (from Arg68 to Lys575) were purified from COS7 cell culture medium using Ni$^{2+}$-affinity chromatography, as described previously[64,89]. For GnT-V activity assay in cultured medium, the media were collected and concentrated with Amicon-Ultra filter units as described above (see *Sample preparation* section). Three µl of concentrated culture medium was directly added to the reaction mixture for the measurement of

GnT-V activity and incubated at 37 °C for 16 h. After enzymatic reactions, samples were heated at 99 °C for 2 min to inactivate enzyme, and 40 µl of water were added. After centrifugation at 21,500 × g for 5 min, the supernatants were analyzed by reverse-phase HPLC with an ODS column (4.6 × 150 mm; TSKgel ODS-80Tm Tosho Bioscience).

**Western and lectin blotting**. The same amount of protein was loaded in each well and proteins were separated by 5%–20% SDS-PAGE followed by transfer to a nitrocellulose membrane. For western blotting, membranes were blocked with 5% skim milk in TBS containing 0.1% Tween-20 (TBS-T) for 30 min, followed by incubation with primary antibody diluted with 5% skim milk/TBS-T overnight at 4 °C. After washing with TBS-T three times, membranes were incubated with secondary antibody conjugated with HRP for 1 h at room temperature. For lectin blotting with biotinylated lectin, membranes were blocked with TBS-T for 30 min, followed by incubation with the biotinylated lectin overnight at 4 °C. After washing with TBS-T three times, membranes were incubated with the contents of a VECTASTAIN ABC kit (1:400) for 1 h at room temperature. For blotting with HRP-conjugated lectin, membranes were blocked with 1% BSA/TBS-T overnight at 4 °C, followed by incubation with HRP-conjugated lectin diluted with 1% BSA/TBS-T. Signals were detected using FUSION SOLO 7 s EDGE (Vilber). Dilutions of antibodies and lectins used were: anti-GnT-V (1:500 and 1:300, from Dr. Miyoshi for endogenous GnT-V in HEK293, CHO, and Neuro-2A cells, or R&D systems, respectively); anti-GAPDH (1:2,000); anti-myc (1:2,000); anti-β-catenin (1:500); anti-SQSTM1/p62 (1:1,000); anti-HA (1:2,000); anti-FLAG (1:500); anti-actin (1:2,000); anti-ZsGreen1 (1:2,000); anti-CANT1 (1:500); anti-B4GALT1 (1:500); Con A-biotin (1:20,000); PHA-L4-HRP (1:20,000); PHA-E4-HRP (1:20,000); DSA-HRP (1:2,000); AAL-biotin (1:2,000); RCA-I-biotin (1:1,000); anti-mouse IgG-HRP (1:10,000); anti-rabbit IgG-HRP (1:20,000); and anti-goat IgG-HRP (1:20,000).

**Immunofluorescence imaging**. Cells were seeded on an eight-well chambered slide and transfected with plasmids if needed the following day. Two days after transfection, cells were washed with PBS twice followed by fixation with 4% paraformaldehyde in PBS at room temperature for 15 min. After washing with PBS twice, cells were blocked and permeabilized with blocking buffer (3% BSA, 0.1% NP-40, in PBS) by incubating at room temperature for 30 min. The cells were then incubated with primary antibody at room temperature for 1 h. After washing with PBS three times, the cells were incubated with fluorescence probe-conjugated secondary antibody and TMR-labeled Halo ligand if necessary at room temperature for 1 h. After washing with PBS three times, the samples were mounted with Prolong Diamond Antifade reagent (ThermoFisher Scientific). Dilutions of antibodies and the Halo ligand concentration used were: anti-GM130 (1:500); rabbit anti-HA (1:500); anti-Golgin97 (1:200); anti-calnexin (1:500); rat anti-HA (1:100); anti-mouse IgG-Alexa488 (1:2,000); anti-rabbit IgG-Alexa488 (1:2,000); anti-mouse IgG-Alexa546 (1:2,000); anti-rat IgG-Alexa546 (1:2,000); and Halo-tag TMR ligand, final 50 nM.

**Fluorescence-activated cell sorting (FACS) analysis**. Cells were washed with PBS twice and collected using cell scrapers, followed by centrifugation at 1400 × g for 3 min. The cells were washed with FACS buffer (1% BSA, 0.1% NaN₃, in PBS) once and stained with fluorescently-labeled lectin in FACS buffer on ice for 15 min. For use of biotinylated lectins, the cells were first incubated with biotinylated lectin in FACS buffer on ice for 15 min, followed by incubation with fluorescently-labeled streptavidin in FACS buffer on ice for 15 min. Data were collected with a FACS Melody cell sorter and analyzed using FlowJo software (BD Biosciences). Dilutions of the lectins/reagents were: GSL-II-FITC (1:200); HPA-Alexa488 (1:200); SNA-FITC (1:200); PNA-biotin (1:200); and streptavidin-Alexa488 (1:200).

**RNA extraction, reverse transcription, and real-time PCR**. Total RNA was extracted from cells seeded on a 6-cm dish with TRI Reagent (Molecular Research Center). Total RNA (1 µg) was reverse-transcribed using the SuperScript IV First-Strand Synthesis System with random hexamers (supplied in the kit). The target cDNAs were amplified using TaqMan Gene Expression Master Mix (Applied Biosystems) and the primers and probes described below. The amplified cDNAs were detected using a CFX Connect Real-Time PCR Detection System (Bio-Rad). The primers and probes were purchased from Applied Biosystems, as follows: Hs00159136_m1 for *MGAT5*, Hs00293370_m1 for *SPPL3*, and Hs99999905_m1 for *GAPDH*. Ribosomal RNA (rRNA) was detected using Ribosomal RNA Control Reagents (VIC probe) purchased from Applied Biosystems (4308329). The mRNA levels of *MGAT5* and *SPPL3* were normalized to those of the rRNA and *GAPDH* levels, respectively.

**N-glycan analysis of cell membrane proteins by LC-ESI-MS**. *N*-Glycans were released from cell membrane proteins[90], labeled with aminoxyTMT6 reagent (ThermoFisher Scientific), and analyzed by LC-ESI MS, in accordance with a previously reported procedure[27] with modifications. Cell pellets (5 × 10⁷ cells) were suspended in 2 mL of homogenization buffer [50 mM Tris-HCl (pH 7.4), 100 mM NaCl, 1 mM EDTA, and protease inhibitor cocktail] by pipetting and then homogenized using a polytron homogenizer, followed by centrifugation at 760 × g for 20 min at 4 °C to remove nuclei and unbroken cells. The supernatant was diluted with 2 mL of Tris-buffer [50 mM Tris-HCl (pH 7.4), 100 mM NaCl] and

then ultracentrifuged at 120,000 × g for 80 min at 4 °C. The membrane pellet was suspended in 100 µL of Tris-buffer, followed by addition of 400 µL of Tris-buffer containing 1% Triton X-114 with pipetting. The lysate was incubated on ice for 10 min and then at 37 °C for 20 min, followed by phase partitioning by centrifugation at 1940 × g for 2 min. The upper (aqueous) phase was removed, and the lower (detergent) phase was mixed with 1 mL of ice-cold acetone and kept at −25 °C overnight. After centrifugation at 1940 × g for 2 min, the precipitated membrane proteins were dissolved in 11 µL of 8 M Urea and spotted (2.5 µL × four times) onto an ethanol-pretreated polyvinylidene fluoride membrane. After drying at room temperature for >4 h, the membrane was washed with ethanol for 1 min once and then with water for 1 min three times. The proteins on the membrane were stained for 5 min with Direct Blue 71 [Sigma Aldrich; 800 µL solution A (0.1% Direct Blue 71) in 10 mL solution B (acetic acid:ethanol:water = 1:4:5 v/v/v)]. After destaining with solution B for 1 min, the membrane was dried at room temperature for >3 h. The protein spots were excised from the membrane and placed into wells of a 96-well plate. The spots were covered with 10 µL of 1% (w/v) poly(-vinylpyrrolidone) 40000 in 50% (v/v) methanol, agitated for 20 min and washed with water (100 µL, five times). PNGase F (2 U in 10 µL of 20 mM phosphate buffer, pH 7.3) was added to the well and the spots were incubated at 37 °C for 15 min, followed by the addition of 10 µL of water and incubation at 37 °C overnight. The samples were sonicated (in the 96-well plate) for 10 min and the released *N*-glycans (20 µL) were transferred to a 1.5 mL polypropylene tube. The wells were washed with water (50 µL, twice), and the washings were added to the tube, and the contents of the tube were evaporated. The dried *N*-glycans were reacted with 200 µL of 2 M acetic acid for 2 h at 80 °C for desialylation. The desialylated *N*-glycans were reacted with aminoxyTMT6 reagent (0.02 mg in 200 µL of 95% methanol, 0.1% acetic solution) by continuous shaking for 15 min at room temperature. After evaporating the reaction solution, 200 µL of 95% methanol was added to the samples, followed by further shaking for 15 min. After evaporating the samples, 100 µL of 10% acetone solution was added to the samples, followed by incubation at room temperature for 15 min with continuous shaking. Excess reagent was removed using Sepharose CL4B resin. The desialylated aminoxyTMT6 glycans were divided into two fractions. One was analyzed by LC-ESI MS as desialylated glycans. The other was treated with β-galactosidase (83 mU in 50 µL of 50 mM ammonium acetate buffer, pH 5.0; Seikagaku Corporation) at 37 °C overnight, boiled for 5 min, and evaporated. The desialylated and degalactosylated glycans were dissolved in 20 µL of 10 mM ammonium bicarbonate solution and analyzed by LC-ESI MS.

*N*-Glycans labeled with aminoxyTMT6 were separated on a carbon column[27,91] (5 µm HyperCarb, 1 mm internal diameter × 100 mm; ThermoFisher Scientific) using an Accela and Vanquish HPLC pump (flow rate 50 µl/min, column oven 40 °C) with a sequence of isocratic and two segmented linear gradients: 0–8 min, 10 mM NH₄HCO₃; 8–38 min, 9%–22.5% (v/v) CH₃CN in 10 mM NH₄HCO₃; 38–73 min, 22.5%–51.75% (v/v) CH₃CN in 10 mM NH₄HCO₃; increased to 81% (v/v) CH₃CN in 10 mM NH₄HCO₃ over 7 min; then re-equilibrated with 10 mM NH₄HCO₃ for 15 min. The eluate was introduced continuously into an ESI source (LTQ Orbitrap XL, ThermoFisher Scientific). MS spectra were obtained in positive-ion mode using an Orbitrap (mass range m/z 800 to 2000; capillary temperature 300 °C; source voltage 4.5 kV; capillary voltage 18 V; tube lens voltage 110 V). For MS/MS analysis, the top three precursor ions were fragmented by HCD using a stepped collision energy (normalized collision energy: 35.0; width: 40.0; the steps: 3; minimum signal required: 10000; isolation width: 4.00; activation time: 100) in an Orbitrap. Monoisotopic masses were assigned to possible monosaccharide compositions using the GlycoMod software tool (mass tolerance for precursor ions ± 0.01 Da, https://web.expasy.org/glycomod/). Xcalibur software ver. 2.2 (ThermoFisher Scientific) was used to show the base peak chromatogram (BPC), extracted ion chromatogram (EIC), and to analyze MS and MS/MS data.

Standard *N*-glycans with the identified GlcNAc linkages were prepared from bovine fetuin *N*-glycans whose structures were already determined[92]. Desialylated and degalactosylated biantennary and tri-antennary aminoxyTMT6-labeled *N*-glycans were prepared from bovine fetuin (F2379-100MG, Sigma) as described above. The *N*-glycans were incubated with purified GnT-III, -IVa, or -V with UDP-GlcNAc as described in the section "*Measurement of glycosyltransferase activities*".

**Statistics and reproducibility**. Statistical analysis was performed using GraphPad Prism 8 software (GraphPad Software, Inc., San Diego, CA, USA). All the experimental numbers (n) in this study mean biological replicates. All experiments except for glycomics analysis were performed at least two independent experiments. For quantification and statistical analyses, data were collected from at least three biological replicates as described in figure legends.

**Reporting summary**. Further information on research design is available in the Nature Research Reporting Summary linked to this article.

# Data availability
LC-MS data of *N*-glycomics in HEK293-SLC35A2-KO cells: GlycoPOST (ID: GPST000216) (URL: https://glycopost.glycosmos.org/preview/205327909613f2acfd783f;

PIN CODE: 7554). Uncropped images of membranes and gels are provided as Supplementary Fig. 11, and a source data file are provided as Supplementary Data 3. All other data are available from the corresponding author.

## Material availability

Reagents generated in this study are available from the corresponding author with a completed Materials Transfer Agreement.

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

## Acknowledgements

We thank Dr. Taroh Kinoshita (Osaka University) for providing SLC35A2-KO HEK293 cells and Dr. Eiji Miyoshi (for Osaka University) for providing anti-GnT-V antibody. We also thank Ms. Chizuko Yonekawa, Ms. Emiko Mori, and Ms. Mayumi Yamada (Gifu University) for technical help. We also thank James Allen, Ph.D., from Edanz Group (https://jp.edanz.com/ac) for editing a draft of this manuscript. This work was partially supported by a Grant-in-Aid for Early-Career Scientists to T.H. [20K15746], an ACT-X grant ([JPMJAX201B] to T.H.) from Japan Science and Technology (JST), a Grant-in-Aid for Scientific Research (B) to Y.K. [20H03207], a Leading Initiative for Excellent Young Researchers (LEADER) project (Y.K.) from the Japan Society for the Promotion of Science (JSPS), a CREST grant ([18070267] to Y.K.) from JST, a grant from the Takeda Science Foundation to Y.K., and a grant from the Tokyo Biochemical Research Foundation to Y.K.

## Author contributions

T.H.: Conceptualization, Investigation, Data analysis, Drafting the manuscript, and Editing the manuscript. M.T.: Investigation, Data analysis. Y.T.: Investigation, Data analysis. M.N.: Investigation, Data analysis. Y.K.: Conceptualization, Supervision, Editing the manuscript, and Acquisition of foundation.

## Competing interests

The authors declare no competing interests.
