## [Peer Review File · Communications Biology]

Reviewers' comments:

Reviewer #1 (Remarks to the Author):

Hirata et al describe an interesting mechanism for regulation of intracellular MGAT5 activity by the ER-Golgi protease SPPL3. Through a series of detailed biochemical and genetic experiments, the authors demonstrate that loss of complex N-glycan formation leads to a reduction of SPPL3 activity against MGAT5, hence increased intracellular accumulation and activity. Furthermore, the authors demonstrate that it is the loss of terminal sialylation that results in reduction of SPPL3-MGAT5 cleavage, rather than other aspects of complex N-glycan formation, such as degree of branching etc. Overall, this adds to our understanding on how N-glycosylation pathways are regulated, with an interesting focus on glycosylation itself as a glycosylation enzyme regulatory mechanism.

I found this to be an interesting study, well conducted, and very well written. I only have a few comments. Otherwise I am happy to support publication.

- P3L131 – Chinese not Chinesese

- P5L214 and Fig 3g – “N110-118S and N334S mutants were secreted at similar levels to WT GnT-V, whereas...” The blot looks like there is reduced secretion of these mutants after PNGase digest? Which blot (PNGase – or +) were the quantitation bar graphs measured from?

o In the same vein, I would like to see an in vitro SPPL-3 assay (as in Fig 3e) done against the N110-118S, N334S, N433S and N447S mutants. This would really complement the blotting data in Fig 3g.

- The use of so many lectin blots could be hard to follow, especially for the glycol-uninitiated. It might be useful to have a diagrammatic reference for the glycan structures recognised by each lectin. Perhaps as part of Fig 1a or else in the SI

- Fig 6a and 6b – where is the cell expression in these blots? Were they done differently to the previous figs? Please clarify.

Reviewer #2 (Remarks to the Author):

Summary

The authors show that by inducing the loss of complex N-glycans, either by use of inhibitors or knocking out glycan modifying enzymes, an increase on both protein level and enzymatic activity of GnT-V is observed. The increase on GnT-V protein level is most likely caused by reduced shedding of its extracellular domain by SPPL3, which is specific to GnTV and mainly dependent on the terminal glycan modifications of GnTV and not on the branching of N-glycans. This is suggested as a novel mechanism of GnT-V activity regulation.

Generally, the work is highly relevant, the findings are novel and interesting for a broad audience, since they may have impact on the development of diseases like for instance cancer and cancer metastasis. Thus, the manuscript is potentially suitable for publication in the Communication Biology Journal. However, some weaknesses must be addressed before final decision. Major issue is the extremely limited number and low quality of experiments addressing the localisation of GnT-V following the inhibition of complex N-glycans, but also the localisation of SPPL3 upon same conditions. As changes in the N-glycans are known to affect the subcellular localisation of proteins and due to the complexity of the Golgi apparatus, more effort needs to be taken to show that the substrate and protease would meet under the altered conditions, especially in regard to the precise localisation of hypoglycosylated GnT-V. Furthermore, the SPPL3 in vitro assay would need to be better controlled and quantified with multiple repeats.

Major points

1. For all experiments it should be mentioned in the manuscript whether the n reflects technical (cells plated on the same day from the same founder) or biological replicates (different founder dishes ideally plated on different days/weeks).
2. Fig. 1c: The authors state that the activities of GnT-IV, GnT-III and Fut-8 in kifunensine-treated cells were almost the same as in untreated cells. However, to me it looks like they are lower, particularly for GnT-IV. This should be commented or phrased more carefully. Also a quantification

of these results would be helpful, to compare the increase in GnT-V activity with the potential activity-loss of the other transferases.

3. Why do the authors switch cellular model systems in one figure? E.g. in Fig. 1i B16 cells are used, while in the other subpanels HEK293 cells are used. This should be explained.

4. Fig. 1i: The precise localisation of GnT-V even within the Golgi, is very crucial at this point to ensure that SPPL3 would still meet the substrate and given the knowledge of how N-glycans affect cellular localisation. Additional and more precise stainings are required. The top cell in Kif-stainings looks particularly bad. Usage of additional markers localised in different areas of the Golgi, specific for example to the trans-Golgi network should be considered. The colocalisation with such markers should be evaluated via colocalisation assays and shown to be unchanged. In addition, the Golgi-localisation of endogenous SPPL3 under kifunensine treatment should be confirmed. Otherwise, it is unclear if SPPL3 does not cleave GnT-V due to the glycan modification or if it rather does not meet it in the Golgi.

5. Fig. 1g: please clarify for this figure and all following figures that this applies to, if the GnT-V relative expression is calculated from the PNGase treated samples or the non-treated.

6. Fig. 2a: By showing that the mRNA of MGAT5 is not changed in GNT-I ko cells, the others conclude that the effects on GnT-V expression occur post-translationally. But how can translational effects be excluded. For instance, GnT-I could affect a regulation factor of translation and, by that, induce enhance GnT-V translation. I'm aware that this is tough to address experimentally, but it should be at least discussed properly.

7. Fig. 2d: WB is not clear enough, presenting a clearer WB is advised.

8. Fig. 2e: GAPDH, the loading control, seems very uneven in lysates (reduced in B16 WT cells) making it hard to assess the accumulation of GnTV in KO cells. Middle and right panel of figure are not mentioned at all in the text.

9. Fig.3c: By showing that expression of overexpressed SPPL3 is not altered in GnT-I deficient cells, the authors conclude that altered N-glycan structures do not affect expression of SPPL3. However, I assume that the expression construct does not contain potential sites of translational regulation, like 3'- or 5'-UTRs, so to my opinion they can not exclude translational regulation of SPPL3 expression by altered N-glycan structures. This should be either addressed or discussed properly.

10. Fig. 3d: Stainings of a better quality and bigger size should be shown. SPPL3 looks like localising mainly in the ER and not the Golgi, possibly an effect of the strong overexpression of SPPL3. Maybe cells with a moderate overexpression of SPPL3 will show a more Golgi localised protein. Co-localisation with ER markers would be very useful. In some of the Kif+ cells, the Golgi staining itself seems affected (right cell, top cell and bottom left cell), is that a side effect from kif treatment? If so, it should be addressed.

11. Fig. 3e: The in vitro assay would need major revisions if it is to be published. Important negative control would include the incubation of an additional aliquot of the purified substrate and protease at 4°C for the same amount of time as the positive sample is incubated at 37°C. No decrease in substrate amount should be observed in the 4°C incubated sample. For more detailed information on similar in vitro assays see Winkler E et al, Biochemistry. 2009 Feb 17;48(6):1183-97. doi: 10.1021/bi801204g. PMID: 1915923.

12. Following the addition of the proper negative control, the in vitro assay should be repeated in at least three different preparations and statistics should be properly calculated and presented to claim a 40% decrease in substrate abundance.

13. Fig. S4: Why was the ko only sequenced in B16 cells, but not in HEK293 cells. It would be good to also show the sequencing for HEK293 cells.

14. Fig. 4: Is there a reason why for the rescue condition, confirmation blots are shown for some of the targets (glycosyltransferase IV, III) but not others (C1, A1, A2)?

15. Fig. 4: By observing no change in the activity of GnT-V the authors conclude that shedding of GnT-V is not changed. A confirmation of this by explicitly showing the shedding e.g. on a Western Blot under these conditions would be helpful.

16. Fig. S6: Why is the effect of C1 ko and A1 ko shown on DNA-level and the effect of A2 Ko by use of a lectin assay? Please comment on the reason for that.

17. In Fig. 4e: is it certain that the difference in GnT-V relative expression between WT and A1 KO is not significant, while in figure 4c (WT vs A2 KO) it is? Error bars are smaller and difference greater in the A1 cells, statistical analysis might need to be revised.

18. Fig. 4k: Based on your Figure 1a, it would seem that the branching performed by GnT-V is crucial for the addition of galactose and sialic acid. However, if that is the case, you would expect

that knocking-out GnT-V would produce the same results as the lack of galactosylation and sialylation so this should either be explained at this part or model from figure 1a should be modified. In addition, it should be mentioned why APP was used in the experiment. Secretion control? Control for O-glycosylation?

19. Fig. 6b: Effect of GnT-I KO on CANT1 secretion should not be minimised in order to claim that "among glycan-related enzymes, GnT-V shedding is selectively regulated...(line 308)" as it is clear from your analysis that the secretion of CANT1 is significantly decreased by half ($p < 0.005$). This could point to a more general mechanism that affects multiple glycosylating enzymes to different extents and should be also discussed.

Minor points

1. Line 34: "of" should be omitted.
2. Line 43: The 14 sugar residues are trimmed before being transported to the Golgi.
3. Line 56: "because" should be changed to "as" since there is not a strong causative connection.
4. Line 60: "of" should be omitted.
5. Line 81: "the" should be removed.
6. Line 84: It is stated that knocking out SPPL3 abolished secretion of GnT-V. To my knowledge it "only" significantly reduced secretion of GnT-V and is thus considered the major sheddase of GnT-V, but it is not excluded that also other proteases contribute to release of GnT-V ectodomain.
7. Line 101: Kifunesine function should be better explained; it does not change the complex glycans back to "immature oligomannose" but rather blocks the removal of the mannoses, which is a necessary step for the formation of complex N-glycans. This would mean that they remain as highmannose type N-glycans.
8. Figure S2d: Increase of GnT-V in PNG treated Lec1 cells compared to WT not visible in this blot. Consider using a different replicate. WT levels are very high, despite the clear decrease when GnT-I is rescued.
9. It is not clear to me, why different statistical post-hoc analysis for the one-way ANOVA was used basically the same statistical problem (for instance in Fig 2 a and c). This should be explained.

Reviewer #1

Hirata et al describe an interesting mechanism for regulation of intracellular MGAT5 activity by the ER-Golgi protease SPPL3. Through a series of detailed biochemical and genetic experiments, the authors demonstrate that loss of complex N-glycan formation leads to a reduction of SPPL3 activity against MGAT5, hence increased intracellular accumulation and activity. Furthermore, the authors demonstrate that it is the loss of terminal sialylation that results in reduction of SPPL3-MGAT5 cleavage, rather than other aspects of complex N-glycan formation, such as degree of branching etc. Overall, this adds to our understanding on how N-glycosylation pathways are regulated, with an interesting focus on glycosylation itself as a glycosylation enzyme regulatory mechanism.

I found this to be an interesting study, well conducted, and very well written. I only have a few comments. Otherwise I am happy to support publication.

Response

Thank you very much for your positive comments and useful suggestions. According to your comments, we revised the manuscript as follows.

Comment 1

P3L131 – Chinese not Chainese

Response

We corrected it, thank you.

Comment 2

P5L214 and Fig 3g – “N110-118S and N334S mutants were secreted at similar levels to WT GnT-V, whereas...” The blot looks like there is reduced secretion of these mutants after PNGase digest? Which blot (PNGase – or +) were the quantitation bar graphs measured from?

Response

Thank you for your comments. We quantified PNGase F-treated blots and calculated the signal ratios (signal in medium / total signal from cell and medium). To clearly describe this point, we revised the figure legend; 'The graph shows quantification of the signal ratio of secreted GnT-V to total (cell + medium) GnT-V in PNGase F-treated blot.' (page 23, line 986-987). Quantification was carried out using 5 independent experiments, and we concluded that secretion of these mutants was similar to WT.

Comment 3

In the same vein, I would like to see an in vitro SPPL-3 assay (as in Fig 3e) done against the N110-118S, N334S, N433S and N447S mutants. This would really complement the blotting data in Fig 3g.

Response

Thank you very much for your insightful suggestion. We agree that the in vitro assay using these N-glycosylation mutants would be ideal, but as responded to Reviewer 2 (response to comment 11 and 12), we repeatedly tried in vitro activity assay of SPPL3 and found that detection of SPPL3 activity in vitro is technically very hard to obtain reliable results. We therefore decided to remove the results of in vitro SPPL3 assays from the present manuscript. Nevertheless, it is surely important to understand the detailed mechanism of how GnT-V cleavage by SPPL3 is blocked by N-glycan alterations, we will tackle this issue in the future as an important next project.

Comment 4

The use of so many lectin blots could be hard to follow, especially for the glycol-uninitiated. It might be useful to have a diagrammatic reference for the glycan structures recognised by each lectin. Perhaps as part of Fig 1a or else in the SI

Response

Thank you very much for your good suggestion. We totally agree that such figure would help non-experts of glycobiology understand what glycan structures are recognized by

the lectins used. According to your suggestion, we added a schematic for the glycan structures recognized by each lectin as a new Fig. S1.

Comment 5

Fig 6a and 6b – where is the cell expression in these blots? Were they done differently to the previous figs? Please clarify.

Response

We appreciate your comments. Basically, the expression levels of GnT-V in cells are very low, while the levels of secreted GnT-V are relatively high and more detectable. Therefore, it is difficult to detect both cellular GnT-V and secreted one at the same time even using high concentrations of the antibody. We have added a long exposed picture to Fig. 6a and replaced the panels of Fig. 6b to another set of experiment. These experiments in Fig. 6b were performed independently to the previous figures.

Reviewer #2

The authors show that by inducing the loss of complex N-glycans, either by use of inhibitors or knocking out glycan modifying enzymes, an increase on both protein level and enzymatic activity of GnT-V is observed. The increase on GnT-V protein level is most likely caused by reduced shedding of its extracellular domain by SPPL3, which is specific to GnTV and mainly dependent on the terminal glycan modifications of GnTV and not on the branching of N-glycans. This is suggested as a novel mechanism of GnT-V activity regulation.

Generally, the work is highly relevant, the findings are novel and interesting for a broad audience, since they may have impact on the development of diseases like for instance cancer and cancer metastasis. Thus, the manuscript is potentially suitable for publication in the Communication Biology Journal. However, some weaknesses must be addressed before final decision. Major issue is the extremely limited number and low quality of experiments addressing the localisation of GnT-V following the inhibition of complex N-glycans, but also the localisation of SPPL3 upon same conditions. As changes in the N-glycans are known to affect the subcellular localisation of proteins and

due to the complexity of the Golgi apparatus, more effort needs to be taken to show that the substrate and protease would meet under the altered conditions, especially in regard to the precise localisation of hypoglycosylated GnT-V. Furthermore, the SPPL3 in vitro assay would need to be better controlled and quantified with multiple repeats.

Response

Thank you very much for your positive comments and critical suggestions. We really appreciate that our paper was evaluated as highly relevant, novel and interesting. According to your comments, we revised the manuscript as follows.

Major points

Comment 1

For all experiments it should be mentioned in the manuscript whether the n reflects technical (cells plated on the same day from the same founder) or biological replicates (different founder dishes ideally plated on different days/weeks).

Response

All experimental numbers (n) in this study mean biological replicates. We added a following sentence in “statistical analysis” section in experimental procedures to show this point. ‘All the experimental numbers (n) in this study mean biological replicates.’ (page 15, line 674)

Comment 2

Fig. 1c: The authors state that the activities of GnT-IV, GnT-III and Fut-8 in kifunensine-treated cells were almost the same as in untreated cells. However, to me it looks like they are lower, particularly for GnT-IV. This should be commented or phrased more carefully. Also a quantification of these results would be helpful, to compare the increase in GnT-V activity with the potential activity-loss of the other transferases.

Response

Thank you for your comments. In Fig. 1c, the overall chromatograms of the untreated cell samples (Kif(-), black line) were shifted up to avoid overlapping of the two baselines, and the activities were measured by calculating the peak areas based on respective baseline. In enzyme reaction, the same amount (100 pmol) of the substrate glycan was added to all the reaction mixtures, and the amount of the remaining substrate is much larger than all the product peaks of GnT-IV, GnT-III, and FUT8, regardless of the kifunensine-treatment. Quantification of the specific activities of the enzymes calculating from the product peak area are shown in Fig. 1d, and this indicates that there are no significant differences between kifunensine-treated and untreated cells for GnT-III, IV and FUT8. To clearly indicate that Fig. 1d shows the quantification of Fig. 1c, we added a following sentence; 'We quantified the activities of these enzymes in the cell lysates by calculating the peak areas of the products' (page 3, line 110-111).

Comment 3

Why do the authors switch cellular model systems in one figure? E.g. in Fig. 1i B16 cells are used, while in the other subpanels HEK293 cells are used. This should be explained.

Response

For most experiments we used HEK293 cells because of technical convenience (efficient transfection and genome editing). Regarding immunofluorescence staining (Fig. 1i and 3d), however, adherence of HEK293 cells is weak and they are easily detached during staining. Therefore, we switched cells to B16 or HeLa cells. This point was included in the revised manuscript as follows, "We used B16 cells for these experiments, because HEK293 cells were easily detached during staining." (page 3, line 138-139).

Regarding Fig. 2, we used B16 cells, because endogenous GnT-V protein is highly expressed and more easily detected in B16 cells than in HEK293 cells. This point has been added in the revised manuscript as follows, "We used B16 cells here, because endogenous GnT-V protein is more easily detected in B16 than in HEK293 cells." (page 4, line 169-170).

Comment 4

Fig. 1i: The precise localisation of GnT-V even within the Golgi, is very crucial at this point to ensure that SPPL3 would still meet the substrate and given the knowledge of how N-glycans affect cellular localisation. Additional and more precise stainings are required. The top cell in Kif- stainings looks particularly bad. Usage of additional markers localised in different areas of the Golgi, specific for example to the trans-Golgi network should be considered. The colocalisation with such markers should be evaluated via colocalisation assays and shown to be unchanged. In addition, the Golgi-localisation of endogenous SPPL3 under kifunensine treatment should be confirmed. Otherwise, it is unclear if SPPL3 does not cleave GnT-V due to the glycan modification or if it rather does not meet it in the Golgi.

Response

We really appreciate your critical comments. We agree that using different organelle markers and analysis of colocalization with markers are critically important to clarify whether the decreased shedding by SPPL3 is due to the altered GnT-V localization or not. According to your comments, we performed co-staining of GnT-V-Halo together with the *cis*- and *trans*-Golgi markers, GM130 and Golgin97, or the ER marker, calnexin, and added the results of line scan analyses. These analyses indicated that fluorescence intensity of GnT-V-Halo was well overlapped with that of GM130 and Golgin97 (new Fig. 1i), particularly with GM130, but not with that of calnexin (new Fig. S5) in cells with or without kifunensine treatment. These results indicated that altered glycan structures did not influence the localization of GnT-V itself. Therefore, we revised the manuscript as follows: 'Co-staining with *cis*- and *trans*-Golgi markers GM130 and Golgin97 indicated that GnT-V-Halo signals were well overlapped with both GM130 and Golgin97 signals, particularly with GM130, in cells with or without kifunensine treatment (Fig. 1i). On the other hand, co-staining with an ER marker Calnexin showed that the pattern of GnT-V-Halo signals was clearly different from that of Calnexin with or without kifunensine treatment (Fig. S5), suggesting that kifunensine-treatment hardly altered the localization of GnT-V-Halo. Taken together, these results demonstrate that loss of complex-type *N*-glycans specifically increased the protein level and activity of GnT-V without affecting its subcellular localization.' (page 4, line 139-146)

In addition, we tried to detect endogenous SPPL3 using commercially available antibodies. Unfortunately, however, even the antibody termed 7F9, which was previously used for the detection of endogenous SPPL3 in HEK293 (Voss et al., 2014, *EMBO J*),

somehow did not work to detect endogenous SPPL3 in B16 cells (above, Fig. A) and HEK293 cells (not shown). We also purchased another antibody from Abcam (#ab251959), but it also did not detect SPPL3 by western blot and immunofluorescence microscopy even after transfection (Fig. B and C). Thus, we instead newly constructed SPPL3 plasmids and expressed 3HA-tagged SPPL3 at weak levels under the control of the thymidine kinase promoter using pTK plasmid. We successfully detected SPPL3-3HA by immunofluorescence microscopy as shown in new Fig. 3d and found that SPPL3-3HA expressed under the weaker promoter is localized in both the ER and Golgi (new Fig. 3d). This result is consistent with a previous report (Krawitz et al., (2005) J. Biol. Chem.), suggesting that SPPL3 is localized in both the ER and Golgi. Together with another point described below (comment 10), we revised the manuscript as follows: 'Moreover, we observed that the localization of SPPL3-3HA was not altered by kifunensine treatment, the protein being mainly localized to the ER and the Golgi (as also previously reported⁴⁸) regardless of the treatment (Fig. 3d), suggesting that SPPL3 encounters GnT-V in kifunensine-treated cells. As expected, co-localization of SPPL3-3HA and GnT-V-Halo was observed in both kifunensine-treated and -untreated cells (Fig. 3e). These results demonstrate that altered N-glycan structures had a negligible effect on the expression and localization of SPPL3.' (page 5, line 201-206).

Comment 5

Fig. 1g: please clarify for this figure and all following figures that this applies to, if the GnT-V relative expression is calculated from the PNGase treated samples or the non-treated.

Response

Thank you for your comment. A similar question was also asked by reviewer #1. We added the statement in each Figure legend to indicate that the quantification was based on the PNGaseF-treated blots (Fig. 1g, 3h, 4c, 4e, and 4g).

Comment 6

Fig. 2a: By showing that the mRNA of MGAT5 is not changed in GNT-I ko cells, the others conclude that the effects on GnT-V expression occur post-translationally. But how can translational effects be excluded. For instance, GnT-I could affect a regulation factor of translation and, by that, induce enhance GnT-V translation. I'm aware that this is tough to address experimentally, but it should be at least discussed properly.

Response

Thank you very much for your important comments. We understand that GnT-I could affect GnT-V translation as well. As GnT-I is known to be localized in the Golgi, we presume that it is less likely that the Golgi protein directly regulates translation events in the rough ER. Furthermore, we have not detected apparent augmentation of the activity of other glycosyltransferases (Fig. 1d). Collectively, we reasoned that GnT-V is posttranslationally upregulated. Accordingly, we revised the manuscript as follows; 'The levels of GnT-V mRNA (encoded by the *MGAT5* gene) were found to be comparable between WT and GnT-I-deficient cells (Fig. 2a), indicating that the level of GnT-V protein is post-transcriptionally upregulated in GnT-I-KO cells. Considering that the other related glycosyltransferases were not upregulated in GnT-I-KO cells (Fig. 1d) and that GnT-I is a Golgi-resident glycosyltransferase and functions in the post-ER organelles, it is less likely that general translational regulation machinery was activated. Therefore, we hypothesized that the level of GnT-V protein is post-translationally upregulated by loss of complex-type *N*-glycans.' (page 4, line 150-157).

Comment 7

Fig. 2d: WB is not clear enough, presenting a clearer WB is advised.

Response

We appreciate your advice. We performed western blotting again and obtained clearer results. We replaced the panels in Fig. 2d with the new ones together with a long exposed panel to show cellular GnT-V.

Comment 8

Fig. 2e: GAPDH, the loading control, seems very uneven in lysates (reduced in B16 WT cells) making it hard to assess the accumulation of GnTV in KO cells. Middle and right panel of figure are not mentioned at all in the text.

Response

Thank you for your comments. We agree that the blot of GAPDH was not clear enough. We replaced the panel set to a new one from another experiment (Fig. 2e, left).

We also revised the main text to mention the middle and right panels as follows; 'Kifunensine treatment of WT cells, which resulted in increased Con A signals and decreased PHA-L4 signals (Fig. 2e, middle and right-hand panels), again significantly increased the level of GnT-V protein and activity in cells (Fig. 2e left-hand panel, first and second lanes, and Fig. 2f), while it decreased the levels of secreted GnT-V (Fig. 2e left-hand panel, fifth and sixth lanes).' (page 4, line 174-178). In addition, as responded to comment 18 below, we also added a statement in the legend for APP blot as a control of secreted protein as follows, "APP was blotted as a positive control of secreted protein⁹³." (page 22, line 963-964).

Comment 9

Fig.3c: By showing that expression of overexpressed SPPL3 is not altered in GnT-I deficient cells, the authors conclude that altered N-glycan structures do not affect expression of SPPL3. However, I assume that the expression construct does not contain potential sites of translational regulation, like 3'- or 5'-UTRs, so to my opinion they can

not exclude translational regulation of SPPL3 expression by altered N-glycan structures. This should be either addressed or discussed properly.

Response

Thank you for pointing it out. It is possible that SPPL3 translation is regulated by N-glycan structures. However, as we addressed above comment 6, we suppose that general translation occurring in the ER was less likely altered by knocking out Golgi-enzyme GnT-I. Accordingly, we added the following sentence; **Although translational regulation of *SPPL3* mRNA through 5'- or 3'-untranslated regions was not examined in our expression system, it is less likely that knocking out the Golgi enzyme GnT-I affects translation of *SPPL3* in the ER.**' (page 5, line 199-201).

Comment 10

Fig. 3d: Stainings of a better quality and bigger size should be shown. SPPL3 looks like localising mainly in the ER and not the Golgi, possibly an effect of the strong overexpression of SPPL3. Maybe cells with a moderate overexpression of SPPL3 will show a more Golgi localised protein. Co-localisation with ER markers would be very useful. In some of the Kif+ cells, the Golgi staining itself seems affected (right cell, top cell and bottom left cell), is that a side effect from kif treatment? If so, it should be addressed.

Response

Thank you very much for the critical comments. As mentioned above (response to comment 4), we re-analyzed SPPL3 localization using weakly expressed SPPL3-3HA, and found that SPPL3-3HA was co-localized with both the ER and Golgi markers even under these conditions (new Fig. 3d). We revised the figures with the magnified images.

Regarding a possible effect by kifunensine, different types of Golgi morphology were observed even in untreated cells, and we think that the Golgi morphology in kifunensine-treated cells was unaffected.

Comment 11

Fig. 3e: The *in vitro* assay would need major revisions if it is to be published. Important negative control would include the incubation of an additional aliquot of the purified substrate and protease at 4°C for the same amount of time as the positive sample is incubated at 37°C. No decrease in substrate amount should be observed in the 4°C incubated sample. For more detailed information on similar *in vitro* assays see Winkler E et al, *Biochemistry*. 2009 Feb 17;48(6):1183-97. doi: 10.1021/bi801204g. PMID: 1915923.

Comment 12

Following the addition of the proper negative control, the *in vitro* assay should be repeated in at least three different preparations and statistics should be properly calculated and presented to claim a 40% decrease in substrate abundance.

Response

We appreciate your critical comments. We totally agree your comments that more convincing data are required for making a conclusion about this point. To address these issues, we tried to repeat *in vitro* assay. Unfortunately, however, we could not obtain reproducible results showing efficient cleavage of GnT-V purified from WT cells. Moreover, to determine the best reaction condition for the *in vitro* assay, we considered the position and the types of tag for SPPL3, and pH of reacting buffer, but neither of them worked well (above, Fig. A for different tag). Because we used membrane fraction as an SPPL3 enzyme source, we suspected that crude membrane fraction might inhibit enzymatic reaction toward GnT-V, resulted in inefficient cleavage of GnT-V *in vitro*. To avoid this problem, we tried to perform similar *in vitro* assay using purified SPPL3 according to the previous literature that you mentioned in your comment (Winkler et al.,

Biochemistry, 2009). To purify SPPL3, we newly constructed expression plasmid encoding SPPL3 C-terminally tagged with Strep tag. However, since expression of SPPL3-Strep tag was not detected well even in WT cell lysates (above, Fig. B), we could not purify SPPL3-Strep and never complete this assay. Therefore, we concluded that *in vitro* SPPL3 activity assay is technically difficult to yield convincing data, and we decided to remove our data of *in vitro* SPPL3 activity assay (old Fig. 3e) from the revised manuscript. Nevertheless, as we have already mentioned in the reply for reviewer#1 (comment 3), it is surely important to understand the detailed mechanism of how GnT-V cleavage by SPPL3 is blocked by *N*-glycan alterations, we will tackle this issue in the future as an important next project.

Comment 13

Fig. S4: Why was the ko only sequenced in B16 cells, but not in HEK293 cells. It would be good to also show the sequencing for HEK293 cells.

Response

In Fig. S6 (previous S4), in both B16 and HEK293 cells, we designed two gRNAs within one exon to remove an internal region and to easily validate genotypes by band shift in PCR. In the case of HEK293 cells, we designed gRNAs to remove 30 bp, and the clear band shift was detected in this KO clone as expected, indicating that gRNAs correctly cleaved both target sites. Therefore, we did not sequence genome DNA in this clone. On the other hand, in the case of B16 cells, we designed gRNAs to remove 28 bp, but a band shift was not clearly observed in this cell clone probably due to the inefficient simultaneous cleavage. To validate complete knockout, we performed DNA sequencing and confirmed that mutations occurred in both alleles.

Comment 14

Fig. 4: Is there a reason why for the rescue condition, confirmation blots are shown for some of the targets (glycosyltransferase IV, III) but not others (C1, A1, A2)?

Response

For the rescue experiments, we used non-tagged constructs for SLC35A1 and SLC35A2 and myc-tagged constructs for GnT-III and GnT-IV. We already knew that myc-tagged GnT-III and IV work well, but we did not know whether C-terminal myc tag affects activity of SLC35A1 and A2 and thus expressed the non-tagged versions, even though we did not have detection antibodies. As functional expression of SLC35A1 and SLC35A2 and the restoration of glycans was clearly confirmed by lectin blotting (Fig. 4c and e), we can assess whether GnT-V expression was reversed by re-expression of these factors.

Comment 15

Fig. 4: By observing no change in the activity of GnT-V the authors conclude that shedding of GnT-V is not changed. A confirmation of this by explicitly showing the shedding e.g. on a Western Blot under these conditions would be helpful.

Response

Thank you very much for pointing it out, and we agree that checking the levels of GnT-V in the media would corroborate our conclusion. We checked the levels of GnT-V protein in cell and media from SLC35C1-KO cells, and found that secretion of GnT-V was not reduced in SLC35C1-KO cells (new Fig. 4c). Therefore, in combination with the activity results, we concluded that fucosylation has no major impact on GnT-V shedding. We revised the manuscript as follows: "We examined cellular GnT-V activity and the secretion of GnT-V, and found that GnT-V activity and the secretion of GnT-V in SLC35C1-KO cells were comparable to those in WT cells (Fig. 4b and 4c), indicating that fucosylation has no effect on GnT-V shedding." (page 5, line 226-229).

Since cellular GnT-V activity is well correlated with the levels of secreted GnT-V in glycosylation-defective cells, we suspect that the level of GnT-V shedding is assessed by the difference in cellular activity. Therefore, we concluded that secretion of GnT-V is hardly affected in GnT-IVa,b-DKO cells as their GnT-V activity is similar to that of WT cells.

Comment 16

Fig. S6: Why is the effect of C1 ko and A1 ko shown on DNA-level and the effect of A2 Ko by use of a lectin assay? Please comment on the reason for that.

Response

Sorry for our poor explanation. We established SLC35C1-KO and SLC35A1-KO cells by ourselves in this study, and the genotypes of these cells were examined during establishing these cell clones. By contrast, SLC35A2-KO cells had already been established in previous study we cited (Y. Wang et al., (2019) J. Biol. Chem.), and simply confirmed the glycan changes. To clearly describe this point, we revised the text as follows; 'We next examined a **previously established** galactosylation-defective HEK293 cell line lacking *SLC35A2*, which encodes a UDP-galactose transporter^{51, 52}. **To validate the glycan changes in SLC35A2-KO cells, we investigated the reactivity of the specific lectins, *Griffonia simplicifolia* lectin II (GSL-II) and *Helix pomation* lectin (HPA), which bind to terminal GlcNAc and *N*-acetylgalactosamine (GalNAc), respectively (Fig. S1)^{53, 54}. Flow cytometry of SLC35A2-KO cells **with GSL-II and HPA** confirmed the loss of galactosylation, and transfection of human *SLC35A2* gene restored the glycan profiles (Fig. S8b).'** (page 5, line 229-235).

Comment 17

In Fig. 4e: is it certain that the difference in GnT-V relative expression between WT and A1 KO is not significant, while in figure 4c (WT vs A2 KO) it is? Error bars are smaller and difference greater in the A1 cells, statistical analysis might need to be revised.

Response

Thank you, and we apologize that the error bars in Fig. 4c and Fig. 4e were mistakenly shown in the previous version (previous Fig. 4c showed SD while Fig. 4e showed SEM). We corrected the graph in Fig. 4e with means \pm SD, and the new graph showed the bigger error bars than those in Fig. 4c. Although statistical significance was not detected in Fig. 4e, every experiment showed the increased tendency of GnT-V protein in A1-KO cells. Furthermore, re-expression of A1 gene reversed the effect, so we concluded that loss of sialylation increased the GnT-V protein level in cells.

Comment 18

Fig. 4k: Based on your Figure 1a, it would seem that the branching performed by GnT-V is crucial for the addition of galactose and sialic acid. However, if that is the case, you would expect that knocking-out GnT-V would produce the same results as the lack of galactosylation and sialylation so this should either be explained at this part or model from figure 1a should be modified. In addition, it should be mentioned why APP was used in the experiment. Secretion control? Control for O-glycosylation?

Response

We appreciate your critical comments. As pointed out, our illustration in previous Fig. 1a was perhaps confusing and made readers misunderstand. Galactosylation and sialylation occur in not only GnT-V-branch but all the branches. Therefore, if GnT-V is lost, galactose and sialic acid are still present in other GlcNAc-branches. To avoid misunderstanding, we modified Fig. 1a.

APP was used as secretion control as you expected. We mentioned this in each legend (Fig. 2e, 3h, and 4k).

Comment 19

Fig. 6b: Effect of GnT-I KO on CANT1 secretion should not be minimised in order to claim that “among glycan-related enzymes, GnT-V shedding is selectively regulated...(line 308)” as it is clear from your analysis that the secretion of CANT1 is significantly decreased by half ($p < 0.005$). This could point to a more general mechanism that affects multiple glycosylating enzymes to different extents and should be also discussed.

Response

Thank you very much for your reasonable comments. We agree that the significant reduction in CANT1 secretion in GnT-I-KO cells should be carefully interpreted and discussed and that *N*-glycan alterations have more broader impacts on secretion. Therefore, we revised the manuscript as follows; ‘In contrast to the severe inhibition of GnT-V shedding in such conditions, secretion of CANT1 was partially inhibited in GnT-I-KO cells, and B4GALT1 secretion was barely affected (Fig. 6b). These results suggest that regulation of

SPPL3-mediated shedding is substrate-dependent, and that shedding of multiple glycan-related enzymes, including GnT-V, are selectively regulated by cellular *N*-glycan structures.' (page 7, line 304-308).

Minor points

Comment 20

Line 34: "of" should be omitted.

Response

We corrected, thank you.

Comment 21

Line 43: The 14 sugar residues are trimmed before being transported to the Golgi.

Response

We revised the sentence as '*N*-Glycosylated proteins are then transported to the Golgi after trimming of several sugar residues,'.

Comment 22

Line 56: "because" should be changed to "as" since there is not a strong causative connection.

Response

We corrected it. Thank you.

Comment 23

Line 60: "of" should be omitted.

Response

We corrected it.

Comment 24

Line 81: “the” should be removed.

Response

We corrected it.

Comment 25

Line 84: It is stated that knocking out SPPL3 abolished secretion of GnT-V. To my knowledge it “only” significantly reduced secretion of GnT-V and is thus considered the major sheddase of GnT-V, but it is not excluded that also other proteases contribute to release of GnT-V ectodomain.

Response

Thank you for your critical comments. We agree with you and rephrased the sentence as ‘Knocking out SPPL3 significantly reduced secretion of GnT-V.’

Comment 26

Line 101: Kifunesine function should be better explained; it does not change the complex glycans back to “immature oligomannose” but rather blocks the removal of the mannoses, which is a necessary step for the formation of complex N-glycans. This would mean that they remain as highmannose type N-glycans.

Response

We agree with you and rephrased the sentence as follow, ‘to inhibit the conversion of immature oligomannose-type *N*-glycans to the complex-type *N*-glycans (Fig. 1a).’

Comment 27

Figure S2d: Increase of GnT-V in PNG treated Lec1 cells compared to WT not visible in this blot. Consider using a different replicate. WT levels are very high, despite the clear decrease when GnT-I is rescued.

Response

Thank you for your suggestion. We replaced the figure with another experimental replicate.

Comment 28

It is not clear to me, why different statistical post-hoc analysis for the one-way ANOVA was used basically the same statistical problem (for instance in Fig 2 a and c). This should be explained.

Response

We used two post-hoc analyses for the one-way ANOVA, Turkey and Dunnett tests. Turkey test compares every mean with every other mean, while Dunnett test compares every mean to a control one, and the detection sensitivity of Turkey is weaker than that of Dunnett. We used Dunnett test when we would like to compare every mean to a control sample as in the case of Fig. 2c. On the other hand, in the case of comparison among all experimental setting (for example, when we would like to compare WT and KO, WT and rescued-cells, and KO and rescued cells as shown in Fig. 2a), we used Turkey test.

Reviewers' comments:

Reviewer #1 (Remarks to the Author):

I have been asked to review this manuscript again. My previous opinion was favourable – I thought this was an interesting study on the effect of crosstalk in N-glycosylation processing.

Overall my opinion is similar to before – I think this is good work and I am happy to support publication. The new revisions mostly add clarity and are welcome. However, there is a bit more cleaning up to do before the paper is ready for publication.

Some minor points:

- L44 "transported to the Golgi after trimming of several sugar residues...".
 - o This phrase isn't very precise. Would be better to specify how many sugar residues. Which sugars etc.

- L47 "Three N-acetylglucosamine (GlcNAc) branches can be generated by N-acetylglucosaminyltransferase (GnT)-III9 , -IV10, and -V6, 11".
 - o Again, this sentence lacks precision and could generate ambiguity. Would be better to specify which 3 branches.

- L71 "In another study, loss of two GlcNAc branches 72 in N-glycans by knocking out GnT-II...".
 - o Lacks precision, which 2?

- L199 "Although translational regulation of SPPL3 mRNA through 5'- or 3'- untranslated regions was not examined in our expression system, but it is less likely that knocking out the Golgi enzyme GnT-I affects translation of SPPL3 in the ER."
 - o I believe this sentence was put in to address concerns raised during the previous round of revisions. However, I think it needs some fixing.
 - o Why is it less likely that ko affects translation of SPPL3?
 - o Also, translation shouldn't be in the ER?

- L226 "We examined cellular GnT-V activity and the secretion of 227 GnT-V, and found that GnT-V activity and the secretion of GnT-V in SLC35C1-KO cells were 228 comparable to those in WT cells (Fig. 4b and 4c)."
 - o Secretion actually seems to go up in C1- KOs?? Fig 4c. Was this checked for significance?

- L258 "In GnT-III-KO cells, the level of GnT-V protein and activity were not increased compared with that in WT cells (Fig. 4h and Fig. 4i)"
 - o This seems supported by the data. However, there also seems to be a slight decrease in GnT-V after GnT-III ko. This should be mentioned and commented upon in the text.

- In general, Fig 4 is very hard to follow. The arrangement of the panels does not follow a logical reading order (i.e. from g>h>i>j>k). This whole figure should be rearranged if possible.

Reviewer #2 (Remarks to the Author):

The authors have addressed almost all concerns. The manuscript has greatly improved and adds important information to the mechanisms of protein glycosylation. Therefore, it is highly relevant and should be published in the Journal of Communications Biology. However, as mentioned in my former comment referring to fig. 4 (Comment 18) A2-KO, A1-KO, IVa,b-DKO and III-KO in Fig. 4 would also profit from analysis of secreted GnT-V. The major take home message (as stated in the title) of the manuscript is that shedding of GnTV is regulated by certain changes in GnT-V glycosylation. So, to my opinion exactly this should be unequivocally demonstrated. I do not agree that it can be assumed that GnT-V secretion in glycosylation-defective cells is necessarily correlated to activity, since the respective knock outs all change the glycosylation status of GnTV which may lead to changes in activity without affecting the shedding.

In addition I have some minor suggestions:

Page 3, line 133 should be changed to:

...which was reversed by GnT-I overexpression (Fig. S3d and S3e).

Page 4, line 154: I-KO cells. Considering that the other related "the" should be omitted.

Page 4, line 154:

Figure 1d does not show GnT-I-Ko cells, but kifunensine treated cells. This should be clarified or corrected.

Page 4, line 155 should read as:

... organelles, it is less likely that loss of GnT-I affected the general translational regulation machinery.

Page 5, line 199f should read as:

Although translational regulation of SPPL3 mRNA through 5'- or 3'- untranslated regions were not examined in our expression system, but (remove the "but") it is less likely that knocking out the Golgi enzyme GnT-I affects translation of SPPL3 in the ER

The new Fig. 4c would profit from statistical analysis, since as depicted now there might be the impression, that secretion of GnTV in C1-KO is slightly increased.

Reviewer #1

I have been asked to review this manuscript again. My previous opinion was favourable – I thought this was an interesting study on the effect of crosstalk in N-glycosylation processing.

Overall my opinion is similar to before – I think this is good work and I am happy to support publication. The new revisions mostly add clarity and are welcome. However, there is a bit more cleaning up to do before the paper is ready for publication.

Thank you very much for reviewing our manuscript again and for your positive comments and important suggestions. We addressed your concerns as described below. We hope that the current manuscript is now ready for publication.

Some minor points:

- L44 “transported to the Golgi after trimming of several sugar residues...”.

o This phrase isn't very precise. Would be better to specify how many sugar residues. Which sugars etc.

We rephrased the sentence as follows: “transported to the Golgi after trimming of **three glucose (and one mannose)** residues,”. (page 1, line 44).

- L47 “Three N-acetylglucosamine (GlcNAc) branches can be generated by N-acetylglucosaminyltransferase (GnT)-III⁹, -IV¹⁰, and -V^{6, 11}”.

o Again, this sentence lacks precision and could generate ambiguity. Would be better to specify which 3 branches.

We fixed the sentence as follows: “**N-Acetylglucosaminyltransferase (GnT)-III⁹, -IV¹⁰, and -V^{6, 11} transfer GlcNAc to β -Man via a β 1,4-linkage, to α 1,3-Man via a β 1,4-linkage, and to α 1,6-Man via a β 1,6-linkage, respectively (Fig. 1a).**” (page 2, line 47-49).

- L71 “In another study, loss of two GlcNAc branches 72 in N-glycans by knocking out GnT-II...”

o Lacks precision, which 2?

We modified the sentence as follows: “In another study, loss of two GlcNAc branches (**β 1,2- and β 1,6-linked GlcNAc residues on α 1,6-Man**) in N-glycans by knocking out

GnT-II led to hyper-*N*-acetylglucosamine extension of the remaining GlcNAc branch for functional compensation in T cells²⁸." (page 2, line 72-75).

- L199 "Although translational regulation of SPPL3 mRNA through 5'- or 3'- untranslated regions was not examined in our expression system, but it is less likely that knocking out the Golgi enzyme GnT-I affects translation of SPPL3 in the ER."

o I believe this sentence was put in to address concerns raised during the previous round of revisions. However, I think it needs some fixing.

o Why is it less likely that ko affects translation of SPPL3?

Given that GnT-I is a glycosyltransferase in the Golgi, knocking out GnT-I likely affects biological processes within Golgi or post-Golgi compartments. Since translation of SPPL3 is carried out in the cytosol and during the ER translocation, it is unlikely that knocking out the Golgi enzyme directly perturbs translation of SPPL3.

o Also, translation shouldn't be in the ER?

Thank you for pointing it out. As SPPL3 is a membrane protein, translation of SPPL3 is initiated in the cytosol. When its hydrophobic transmembrane segment appears, the ribosome is targeted to the ER and the translation is completed.

To include these points, we modified the sentence as follows: "Although translational regulation of *SPPL3* mRNA through 5'- or 3'- untranslated regions was not examined in our expression system, it is less likely that knocking out the Golgi enzyme GnT-I **directly** affects translation of SPPL3 in the ER, **because knocking out GnT-I likely affects biological processes within the Golgi or in post-Golgi compartments**." (page 5, line 201-204).

- L226 "We examined cellular GnT-V activity and the secretion of 227 GnT-V, and found that GnT-V activity and the secretion of GnT-V in SLC35C1-KO cells were 228 comparable to those in WT cells (Fig. 4b and 4c)."

o Secretion actually seems to go up in C1- KOs?? Fig 4c. Was this checked for significance?

Thank you very much for pointing out this point. We performed statistical test and found that there is no significant difference between WT and C1-KO cells. Therefore,

the statement that the secretion of GnT-V in SLC35C1-KO cells were comparable to those in WT cells was validated. We revised Fig. 4c and the legend to clarify that we tested the statistical significance.

- L258 “In GnT-III-KO cells, the level of GnT-V protein and activity were not increased compared with that in WT cells (Fig. 4h and Fig. 4i)”

o This seems supported by the data. However, there also seems to be a slight decrease in GnT-V after GnT-III ko. This should be mentioned and commented upon in the text.

Thank you very much for your critical comments. We agree that a slight decrease of GnT-V activity in GnT-III-KO cells is important, because this may account for a slight increase of GnT-V secretion considering that loss of bisecting GlcNAc increases the number of GlcNAc-branches. We have actually mentioned this point in the discussion part as follows: “Furthermore, we revealed that the protein level and the cellular activity of GnT-V were slightly decreased in GnT-III-KO cells compared with those in WT cells (Fig. 4i and 4j), suggesting that loss of bisecting GlcNAc promotes GnT-V shedding. Because the presence of bisecting GlcNAc suppresses the reactions of various glycosyltransferases, including GnT-IV and GnT-V²⁷, loss of bisecting GlcNAc likely resulted in elevation of the number of GlcNAc branches, which could in turn enhance GnT-V cleavage by SPPL3.”. (page 9, line 388-393).

- In general, Fig 4 is very hard to follow. The arrangement of the panels does not follow a logical reading order (i.e. from g>h>i>j>k). This whole figure should be rearranged if possible.

Thank you very much for your useful comments. Since we added the additional data as mentioned below, we rearranged the order of the figure. We hope that the arrangement of new Figure 4 will be acceptable now.

Reviewer #2 (Remarks to the Author):

The authors have addressed almost all concerns. The manuscript has greatly improved and adds important information to the mechanisms of protein glycosylation. Therefore, it is highly relevant and should be published in the Journal of Communications Biology.

However, as mentioned in my former comment referring to fig. 4 (Comment 18) A2-KO, A1-KO, IVa,b-DKO and III-KO in Fig. 4 would also profit from analysis of secreted GnT-V. The major take home message (as stated in the title) of the manuscript is that shedding of GnTV is regulated by certain changes in GnT-V glycosylation. So, to my opinion exactly this should be unequivocally demonstrated. I do not agree that it can be assumed that GnT-V secretion in glycosylation-defective cells is necessarily correlated to activity, since the respective knock outs all change the glycosylation status of GnTV which may lead to changes in activity without affecting the shedding.

Thank you very much for reviewing our manuscript again and for your positive comments and insightful suggestions. In order to clarify your point, we measured GnT-V activity in cells and culture media, and calculated the ratio. We found that GnT-V activity in culture medium was greatly decreased in A1-KO and A2-KO cells than in WT cells (new Fig. 4h). Thus, we concluded that loss of sialylation inhibits GnT-V secretion through proteolytic cleavage by SPPL3. We added the following sentence:

“Furthermore, GnT-V activity in culture medium was robustly decreased in SLC35A1-KO and SLC35A2-KO cells than in WT cells (Fig. 4h).”. (page 6, line 252-254).

In addition I have some minor suggestions:

Page 3, line 133 should be changed to:

...which was reversed by GnT-I overexpression (Fig. S3d and S3e).

We fixed it. Thank you.

Page 4, line 154: I-KO cells. Considering that the other related

“the” should be omitted.

As described in the next comment, we deleted the phrase “that the other related glycosyltransferases were not upregulated in GnT-I-KO cells (Fig. 1d) and”. Thank you very much for your comment.

Page 4, line 154:

Figure 1d does not show GnT-I-Ko cells, but kifunensine treated cells. This should be clarified or corrected.

Thank you very much for pointing it out. Fig. 1d is actually the case of kifunensine

treated cells as you mentioned. Therefore, we deleted the phrase “that the other related glycosyltransferases were not upregulated in GnT-I-KO cells (Fig. 1d) and”.

Page 4, line 155 should read as:

... organelles, it is less likely that loss of GnT-I affected the general translational regulation machinery.

We corrected it.

Page 5, line 199f should read as:

Although translational regulation of SPPL3 mRNA through 5'- or 3'- untranslated regions were not examined in our expression system, but (remove the “but”) it is less likely that knocking out the Golgi enzyme GnT-I affects translation of SPPL3 in the ER
We removed “but”. Thank you.

The new Fig. 4c would profit from statistical analysis, since as depicted now there might be the impression, that secretion of GnTV in C1-KO is slightly increased.

As described above, we performed statistical test and found that there is no significant difference between WT and C1-KO cells. We revised Fig. 4c and the legend to clarify that we tested the statistical significance.